# Kriging meta-models for damage equivalent load assessment of idling offshore wind turbines

**Franziska Schmidt**[1]**, Clemens Hübler**[2]**, and Raimund Rolfes**[1]

[1]Leibniz University Hannover, Institute of Structural Analysis, ForWind, Appelstr. 9A,
30167 Hanover, Germany
[2]TU Darmstadt, Institute of Structural Mechanics and Design, Franziska-Braun-Str. 3,
64287 Darmstadt, Germany

**Correspondence:** Franziska Schmidt (f.schmidt@isd.uni-hannover.de)

**Abstract.** The use of meta-models (e.g. Kriging, artificial neural networks, and polynomial chaos expansion) as surrogate models of aeroelastic simulation models offers a good opportunity to perform lifetime calculations with a feasible computational effort. Meta-models for the approximation of fatigue loads, i.e. damage equivalent loads, of wind turbines in normal operation have been researched comprehensively in recent years. For offshore wind turbines in particular, however, downtimes, i.e. the times when wind turbines idle, also have a significant impact the lifetime. Currently, there are no meta-models of idling wind turbines available. However, it cannot simply be assumed that the findings from normal operation can be directly transferred to idling, as the structural behaviour differs significantly from normal operation due to the lack of aerodynamic damping and the resulting larger impact of the wave loads. For this reason, for the first time, the creation of meta-models, more precisely Kriging meta-models, for an idling offshore wind turbine is investigated comprehensively in this paper. The investigation of meta-modelling shows that for the approximation of the rotor blade root bending moments, two additional input parameters have to be considered in addition to the input parameters that are used for the creation of a meta-model for same offshore wind turbine in normal operation. The comprehensive investigation of the Kriging meta-models shows that the meta-models trained with 2500 data points represent the simulation model with an acceptable approximation quality when choosing suitable Kriging settings.

## 1 Introduction

To determine the lifetime extension potential of offshore wind turbines, a lifetime recalculation or lifetime reassessment must be carried out. However, lifetime reassessments for offshore wind turbines are very time-consuming, as many combinations of environmental parameters have to be calculated. Due to this high number of input parameter combinations, an exact lifetime reassessment, in which all actually occurring combinations of the environmental parameters are considered, is not possible when using aeroelastic simulation codes. According to IEC 61400-3 (IEC, 2019b), representative combinations of the environmental parameters are therefore calculated, although the number of required simulations

is still very high (approximately 1 000 000 simulations, depending on the site). One way to significantly decrease the computing time for a lifetime reassessment is to lump the combinations of the environmental parameters into a few representative combinations, e.g. Ziegler and Muskulus (2016), Bouty et al. (2017), and Velarde et al. (2020). In Ziegler and Muskulus (2016) and Bouty et al. (2017), the combinations of the environmental parameters are lumped together in such a way that the representative combinations generate the same damage as if all combinations resulting from the scatter diagram had been simulated. The calculated damage is then extrapolated to the lifetime. One drawback of this method is that not all combinations of environmental parameters that occur in reality can be considered. An alternative to a re-

duction in the number of considered input parameter combinations for lifetime calculations is the use of meta-models as surrogate models for the aeroelastic simulation code (e.g. Dimitrov et al., 2018; Schmidt et al., 2023). For this purpose, the meta-models must first be created, and it is important to ensure that the meta-models approximate the aeroelastic simulation model with sufficient accuracy.

In recent years, a lot of research has been done in the area of meta-modelling for the fatigue load prediction of operating wind turbines (onshore and offshore). Investigations have been carried out involving, among other approaches, artificial neural networks (ANNs; e.g. Müller et al., 2017; Haghi and Crawford, 2024), polynomial chaos expansion (PCE; e.g. Murcia, 2018), Kriging or Gaussian process regressions (e.g. Wilkie, 2020; Avendaño-Valencia et al., 2021; Müller et al., 2022), and mixture density networks (e.g. Singh et al., 2024). Comparative studies have also been carried out, for example by Dimitrov et al. (2018), who compared five different meta-models, including Kriging and PCE, for the approximation of fatigue loads, i.e. damage equivalent loads (DELs), of an onshore wind turbine. In addition, Schröder et al. (2018) compared the use of PCE and ANN for the same onshore wind turbine, and Slot et al. (2020) analysed PCE and Kriging for another onshore wind turbine. Müller et al. (2021) compared ANN and Kriging for an offshore wind turbine, and Singh et al. (2024) conducted a comparison of mixture density networks and Kriging for onshore and offshore wind turbines. It was found that for both onshore and offshore wind turbines in normal operation, meta-models for approximating the fatigue loads can provide a suitable alternative to the original aeroelastic simulation model.

However, according to IEC 61400-1 (IEC, 2019a), idling must also be taken into account in addition to normal operation when calculating the lifetime of an offshore wind turbine. Here, the rotor blades are pitched out, and the rotor is usually not braked, so very slow rotation of the rotor is possible. Due to the low rotational speed, there is almost no aerodynamic damping. For onshore wind turbines and small offshore wind turbines, idling has a positive effect on their lifetime due to the lower fatigue loads that occur (Santos et al., 2022; Ziegler et al., 2024). In contrast, the loads of offshore wind turbines can be the same or even higher for idling conditions than for normal operation. This can be explained by the fact that wave loads have a higher impact on structural behaviour due to the lack of aerodynamic damping. As a result, the structure is excited more strongly, which can lead to loads that are not negligibly low compared to the loads during normal operation. For large wind turbines on monopiles in particular, this is relevant as the impact of the wave loads is larger compared to wind turbines on smaller monopiles (Velarde et al., 2020; Santos et al., 2022).

The number of idling conditions is significantly smaller compared to the number of normal operating conditions. However, extrapolated over the lifetime, there is still a large number of idling conditions that needs to be calculated. Ap-

proximately 5 % of the operational lifetime is spent under idling conditions caused by wind speeds below the cut-in or above the cut-off wind speed (here based on the example of the wind speed distribution of FINO 3 (Hübler et al., 2017b) using the NREL 5 MW reference turbine). In addition, wind turbines idle in around 4 %–10 % of their total operational lifetime due to downtimes caused by the general availability of the wind turbine, which are assumed to be uncorrelated with environmental conditions (Horn and Leira, 2019). Adding these together, around 9 %–15 % of their total operational lifetime is spent under idling conditions. Extrapolated to a lifetime of 25 years, this means that more than 100 000 simulations (with a simulation time of 10 min each) must be carried out just to take the idling conditions into account if all combinations of environmental parameters that actually occurred are taken into account. Due to this high number of simulations, a calculation in the time domain, i.e. with aeroelastic simulation codes, is hardly possible. In order to be able to take all these 10 min realisations into account in the lifetime calculation, meta-models must also be created for an idling wind turbine. Nevertheless, to the authors' knowledge, there has been no research so far regarding meta-modelling for idling wind turbines, although idling can have a significant influence on the lifetime of offshore wind turbines, as described above. Due to the lack of aerodynamic damping and the resulting increased impact of wave loads, idling wind turbines have a different structural behaviour compared to turbines in normal operation. For this reason, it cannot be assumed that the findings of previous investigations into meta-modelling of offshore wind turbines in normal operation can be transferred to idling offshore wind turbines. For this reason, in this work, the creation of meta-models, more precisely Kriging meta-models, of an idling offshore wind turbine is investigated comprehensively for the first time. Kriging, also known as Gaussian process regression, was selected because, on the one hand, Kriging meta-models have shown promising results in previous work, in terms of approximation quality and the computational time required to create the meta-models for wind turbines in operating conditions, e.g. Slot et al. (2020); Wilkie (2020). On the other hand, Kriging meta-models already exist for the same simulation model used in this work for normal operation with a good approximation quality (Müller et al., 2022).

To create the meta-models, a database (i.e. training and test data) must first be generated. Due to the almost-nonexistent aerodynamic damping, it can be assumed that the time period in which the initial transients affect the simulation results due to the abrupt loading on the turbine at the start of the simulation is longer than for an operating wind turbine (Hübler et al., 2017b). For offshore wind turbines in normal operation, the time periods for the initial transients, in the following referred to as run-in times, used in the literature vary substantially, generally ranging from 20 s to several hundred seconds, e.g. Jonkman and Musial (2010), Vemula et al. (2010), Müller et al. (2018), and Singh et al. (2024). This is, among

other causes, due to the fact that the run-in times depend on the simulation model (wind turbine type and substructure) and the aeroelastic code used. There are only a few detailed studies in which the required run-in times have been determined. A first detailed study on run-in times was carried out by Zwick and Muskulus (2015) for an offshore wind turbine on a jacket. In this study, however, only normal operation was considered, while idling was not taken into account. Hübler et al. (2017b) also conducted a study of the run-in times of offshore wind turbines for a monopile and a jacket foundation. Here, in addition to normal operation, the run-in times were also analysed for idling conditions for wind speeds below cut-in and above cut-out wind speeds. However, for the idling condition, the run-in times for wind speeds between the cut-in and cut-off wind speeds were not analysed. Due to the very high maximum run-in time of 720 s in Hübler et al. (2017b), it is not reasonable to take this as a conservative assumption for all wind speeds, as this dramatically increases the computational effort. For this reason, before starting meta-modelling, the first step is to determine the run-in time for idling as a function of the wind speed.

The paper is structured as follows. In the first part, the focus is on the time domain simulation of the idling operating state. Here, the run-in times are determined with respect to the wind speed. Then, the next step is to create the training and test data sets for meta-modelling. These are then used to investigate the creation of the meta-models for the idling wind turbine. The Kriging meta-models are analysed with regard to the Kriging settings, i.e. the choice of the covariance and basis function. Subsequently, a convergence study is carried out to determine the number of simulations required for creating the meta-models. Finally, the results of the meta-modelling of the idling offshore wind turbine are compared with the results of the meta-modelling of the same offshore wind turbine in normal operation, and a conclusion is drawn.

## 2 Time domain simulations of an idling offshore wind turbine

### 2.1 Simulation model and environmental conditions

The time domain simulations for the investigations in this work are conducted using the aero-hydro-servo-elastic code FASTv8 from the National Renewable Energy Laboratory (Jonkman, 2013). A soil model by Häfele et al. (2016) and Hübler et al. (2018) that enhances the FASTv8 code is considered. The NREL 5 MW reference turbine (Jonkman et al., 2009) with the OC3 monopile and soil (Jonkman and Musial, 2010) is used as the wind turbine model. For the soil model, the required soil matrices are based on the lateral soil model of Kallehave (2012) and on the axial soil model of FUGRO (API, 2007). These choices follow the work of Hübler et al. (2017a, 2018). In accordance with Häfele et al. (2016), the soil matrices are calculated using initial conditions; i.e. no loads were applied.

The turbulent wind field is calculated using TurbSim (Jonkman, 2016). Here, the Kaimal turbulence model is used, which is one of the turbulence models recommended in the IEC 61400-1 guideline (IEC, 2019a) and which is frequently used, for example by Yang et al. (2015), Slot et al. (2020), and Wilkie (2020). The irregular waves are calculated using the JONSWAP spectrum. The JONSWAP spectrum is commonly used for offshore wind turbine simulations, e.g. by Velarde et al. (2019), Stieng and Muskulus (2020), and Wilkie (2020).

Five scattering parameters (mean wind speed $v_s$, turbulence intensity TI, significant wave height $H_s$, wave peak period $T_p$, and wind–wave misalignment $\theta_{mis}$) are considered in this work. For these five parameters, the statistical distributions by Hübler et al. (2017b) are used, which were fitted to the measured environmental conditions at the FINO 3 research platform in the German North Sea.

The simulation length of each simulation is 10 min in addition to the run-in times to be determined in the next section.

### 2.2 Determination of the run-in times

Before creating the database for the training and testing of the meta-models, the first step is the determination of the run-in times for the NREL 5 MW reference turbine on the OC3 monopile in idling conditions using the FASTv8 software. Since previous investigations have shown that run-in times depend on wind speed (Zwick and Muskulus, 2015; Hübler et al., 2017b), the run-in times are determined as a function of the wind speed.

The run-in times are determined by investigating the internal forces at two representative locations at the wind turbine, which are visualised in Fig. 1. The internal forces of the monopile are investigated at the mud line, and the internal forces of the rotor blade are investigated at its root. An overview of the considered internal forces can be found in Table 1. During the course of this paper, the meta-models are created and analysed for these internal forces.

To obtain the run-in times as a function of the wind speed, wind speeds of 1 to $33\,\mathrm{m\,s^{-1}}$ with a step width of $2\,\mathrm{m\,s^{-1}}$ are considered. A total of 10 000 simulations are carried out for each wind speed. In each of the 10 000 simulations, the random wind and wave seeds and the initial azimuth angle $\psi$ (initial position of the rotor blades) are varied. All other scattering parameters, $H_s$, $T_p$, TI, etc., are kept constant, whereby the mean values of the corresponding probability distribution are used. This means that the 10 000 simulations conducted for every wind speed are identical except for the values of the random wind and wave seeds and the initial azimuth angle $\psi$.

A convergence study is conducted to determine the run-in times. Each of the 10 000 simulations per wind speed has a simulation length of 60 min. This leads to a maximum of 50 min for the initial transients as the usable simulation length is 10 min. Based on the study of Hübler et al. (2017b)

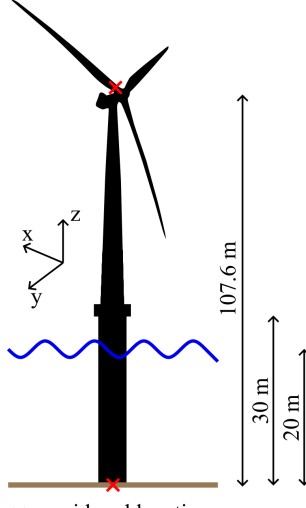

107.6 m

30 m

20 m

× considered locations

**Figure 1.** Visualisation of the NREL 5 MW reference turbine on the OC3 monopile (not to scale) with markings indicating the locations at which the internal forces are evaluated.

**Table 1.** Considered internal forces for the determination of run-in times and for the creation of meta-models.

| Internal forces | Description |
| --- | --- |
| $F_x$ | shear force at mud line in wind direction |
| $M_y$ | overturning moment at mud line in wind direction |
| $F_y$ | shear force at mud line perpendicular to wind direction |
| $M_x$ | overturning moment at mud line perpendicular to wind direction |
| $M_{x,\text{Root}}$ | edgewise moment at the blade root |
| $M_{y,\text{Root}}$ | flapwise moment at the blade root |

on initial transients of an offshore wind turbine, it is assumed that this duration is sufficient for an idling wind turbine.

For the convergence study, the run-in time is increased step by step in 10 s intervals, starting with a minimum run-in time of 10 s. Then, in each step, for each of the 10 000 simulations, the run-in time is cut off, and a short-term DEL is calculated for a simulation length of 10 min according to Eq. (1).

$$S_{\text{eq}} = \left( \sum \frac{n_i S_i^m}{N_{\text{ref}}} \right)^{\frac{1}{m}} \tag{1}$$

Here, $n_i$ is the corresponding number of cycles for each load amplitude $S_i$ determined by rainflow counting according to the ASTM E1049-85 standard (ASTM, 2017). $N_{\text{ref}} = 600$ is the number of equivalent cycles for a frequency of 1 Hz for a 10 min time series, and $m$ is the Wöhler exponent. The Wöhler exponent $m = 3$ is chosen for steel and $m = 10$ for the

composite material of the rotor blade. The load range $S_i$ is corrected according to Goodman (1914).

The resulting 10 000 short-term DELs $S_{\text{eq}}$ are then averaged for each step of the convergence study. The run-in time is considered sufficiently long if the deviation of the current mean short-term DEL from the mean short-term DEL of the maximum analysed run-in time of 50 min is less than 5 %.

As described above, the 10 000 simulations conducted for every wind speed are identical except for the values of the random wind and wave seeds and $\psi$. $\psi$ is varied in each simulation in addition to the random wind and wave seeds, as during the investigations, it turned out that the run-in times, especially for the flapwise bending moment at the blade root $M_{y,\text{Root}}$, could only be determined if $\psi$ is varied in each simulation. This can be explained as follows. In the case of an idling wind turbine, the blades are pitched out. Due to this pitch angle and the very low rotational speed of the rotor, the flapwise (idling: in-plane) bending moment is significantly affected by the weight of the rotor blade itself and therefore depends on the position of the rotor blade. This dependency is shown in Fig. 2, where the time series of the azimuth angle and $M_{y,\text{Root}}$ are shown for one example simulation with a mean wind speed of $v_s = 17 \, \text{m s}^{-1}$. It becomes clear that the value of $M_{y,\text{Root}}$ is significantly influenced by the value of the azimuth angle. Furthermore, it is noticeable that the values of $M_{y,\text{Root}}$ only change very slowly. This is due to the fact that the azimuth angle, i.e. the rotor position, only changes very slowly, caused by the low rotor speed (approximately one rotation per hour). If only the random wind and wave seeds are varied, while $\psi$ has the same value in all 10 000 simulations, the resulting time series of $M_{y,\text{Root}}$ differ only due to different rotor speeds caused by different stochastic wind fields. These differences are – especially for $M_{y,\text{Root}}$ – not large enough to determine a convergence of the run-in times, as the time series are too similar, and the mean values of $M_{y,\text{Root,DEL}}$ are therefore too strongly affected by the rotation of the rotor. This issue can be avoided by varying $\psi$ in each simulation. Due to the different initial rotor positions, the time series of $M_{y,\text{Root}}$ are more different. Consequently, the influence of the rotation of the rotor in $M_{y,\text{Root,DEL}}$ is averaged out when calculating the mean value of $M_{y,\text{Root,DEL}}$. This enables the run-in times to be determined. For this reason, the azimuth angle is varied randomly to determine the run-in times.

Figure 3 shows an example of the convergence behaviour for wind speeds from 1 to 9 m s$^{-1}$ for $M_{y,\text{DEL}}$ and $M_{y,\text{Root,DEL}}$. For the remaining internal forces and wind speeds considered, the convergence behaviour is shown in Appendix A in Figs. A1 to A3. It is clear that $M_{y,\text{DEL}}$ has a better convergence behaviour than $M_{y,\text{Root,DEL}}$ and that longer run-in times are required when looking at $M_{y,\text{Root,DEL}}$. For $M_{y,\text{DEL}}$, it can be seen that the run-in time depends on the wind speed. The higher the wind speed, the shorter the run-in time required. This dependence of the run-in time on the wind speed can be observed for all analysed

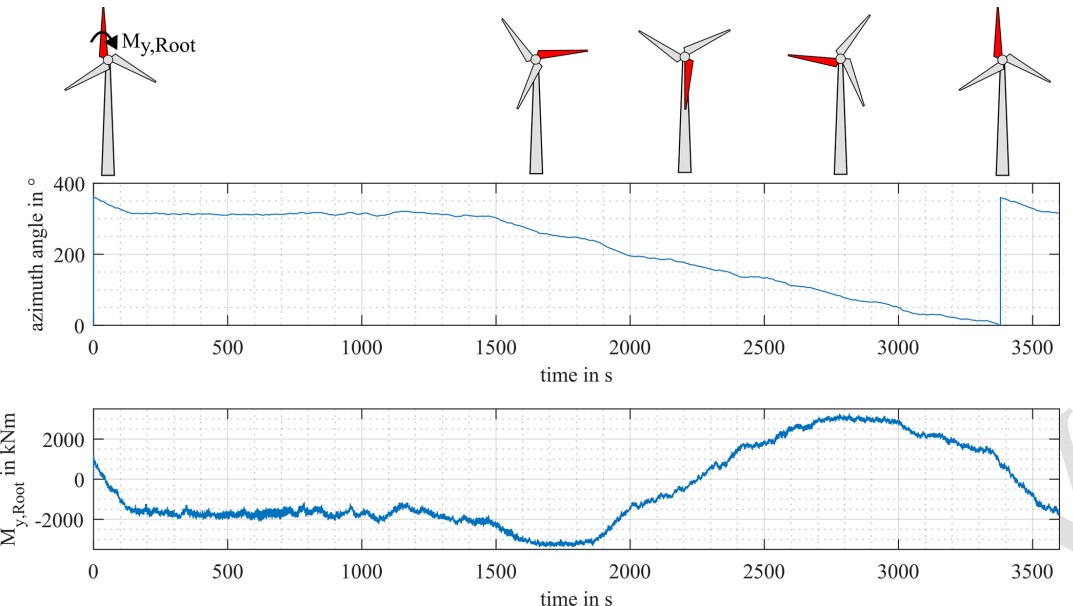

**Figure 2.** Influence of the azimuth angle on the flapwise bending moment $M_{y,\text{Root}}$ at the rotor blade root for a mean wind speed of $v_s = 17\,\text{m}\,\text{s}^{-1}$.

internal forces, although the run-in times of the bending moments at the rotor blade roots deviate from this in some cases. For $M_{y,\text{Root,DEL}}$, for example, it can be seen that the results for 3 and $5\,\text{m}\,\text{s}^{-1}$ deviate from this trend, as shown in Fig. 3. The reason for these deviations could not be determined in this study. As the simulation of $M_{y,\text{Root,DEL}}$ additionally requires the highest run-in times for most wind speeds, $800\,\text{s}$ run-in time is assumed for wind speeds $< 16\,\text{m}\,\text{s}^{-1}$ to be on the conservative side. For very high wind speeds, the investigations show required run-in times of less than $10\,\text{s}$. Here, to be on the safe side, a minimum run-in time of $60\,\text{s}$ is specified. The run-in times defined in this work are shown in Table 2.

Comparing the results in Table 2 with the results of Hübler et al. (2017b) for the same wind turbine in operation, it is noticeable that the trend that the required run-in time becomes shorter with increasing wind speed was observed previously. The results also agree well for wind speeds of less than $3\,\text{m}\,\text{s}^{-1}$, where the turbine idles. However, the results differ significantly in one aspect. Compared to $60\,\text{s}$ in this study, a very long run-in time of $360\,\text{s}$ was determined for the idling wind turbine for wind speeds greater than $25\,\text{m}\,\text{s}^{-1}$. One possible reason for this could be that in this work only one combination of the environmental parameters per wind speed is calculated 10 000 times with different random seeds and varying initial azimuth angles. Here, the mean values of the probability distributions of the remaining scattering parameters are used. The determined run-in times therefore apply to average combinations of environmental parameters. However, rarely occurring combinations of input parameters, which may lead to higher run-in times, are not taken into ac-

count. In Hübler et al. (2017b), in contrast, 10 000 different combinations of environmental parameters per wind speed bin were calculated to determine the run-in times. This means that the wind speed and other scattering parameters, such as turbulence intensity, significant wave height, and wave peak period, were varied in each of the 10 000 simulated 10 min realisations per wind speed bin. As a result, less frequently occurring combinations of input parameters were also taken into account, which may lead to higher run-in times. To summarise, it can therefore be concluded that the run-in times in this work apply to average input parameter combinations, while the run-in times from Hübler et al. (2017b) can be used to be on the conservative side.

## 3 Meta-modelling

### 3.1 Kriging meta-model

In this work, Kriging is used as a meta-model for the simulation model of the offshore wind turbine in idling conditions. Kriging uses a combination of a regression equation to model the mean or general trend in the data and a Gaussian process with a zero mean to model the deviations from the general trend (Santner et al., 2018). The equation of Kriging is given as follows by Rasmussen and Williams (2006):

$$g(\mathbf{x}) = f(\mathbf{x}) + \mathbf{h}(\mathbf{x})^T \boldsymbol{\beta}. \tag{2}$$

$\mathbf{h}(\mathbf{x})^T \boldsymbol{\beta}$ represents the general trend in the data with known regression or basis functions $\mathbf{h}(\mathbf{x})$ and unknown regression coefficients $\boldsymbol{\beta}$. $f(\mathbf{x})$ is a Gaussian process with a zero

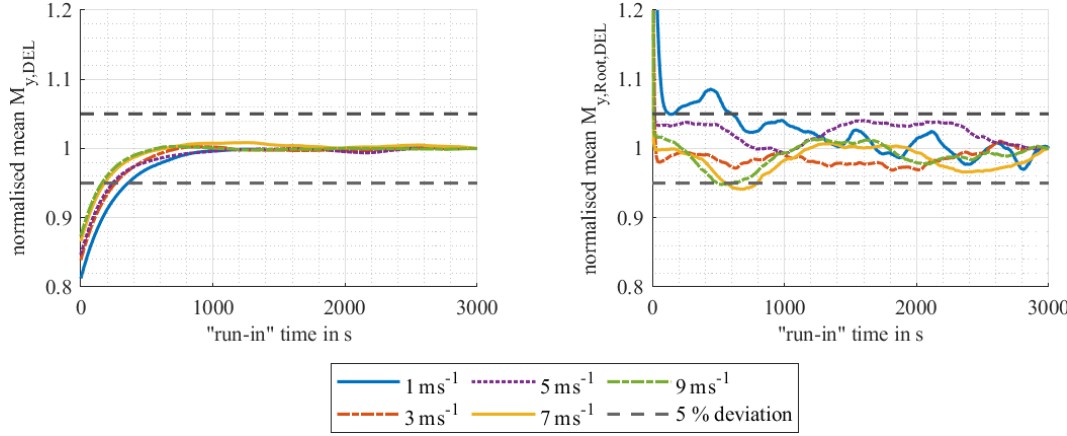

**Figure 3.** Initial transient behaviour of the idling wind turbine for mean wind speeds of 1 to $9\,\mathrm{m\,s^{-1}}$.

**Table 2.** Recommended run-in times that should be discarded to exclude the effects of initial transients.

| $v_s$ in m s$^{-1}$ | < 2 | 2–4 | 4–6 | 6–8 | 8–10 | 10–12 | 12–14 | 14–16 | 16–18 |
|---|---|---|---|---|---|---|---|---|---|
| Run-in time in s | 800 | 800 | 800 | 800 | 800 | 800 | 800 | 800 | 100 |
| $v_s$ in m s$^{-1}$ | 18–20 | 20–22 | 22–24 | 24–26 | 26–28 | 28–30 | 30–32 | > 32 | |
| Run-in time in s | 70 | 70 | 60 | 60 | 60 | 60 | 60 | 60 | |

mean and the covariance function or kernel function $k(\mathbf{x}, \mathbf{x}')$:

$$f(\mathbf{x}) \sim GP(0, k(\mathbf{x}, \mathbf{x}')). \tag{3}$$

As mentioned before, using the Gaussian process (GP), the deviations from the general trend, also called residuals, are modelled. Here, the covariance function $k(\mathbf{x}, \mathbf{x}')$ describes the similarity between different data points ($\mathbf{x}$ and $\mathbf{x}'$). It is assumed that data points whose input values are close to each other, i.e. are similar, also have similar output values. Covariance functions can be isotropic or anisotropic. They differ in the correlation length used. The correlation length indicates how distant data points may be from each other in order for them to still be correlated. For isotropic covariance functions, there is a joint correlation length for all input parameters. In contrast, for anisotropic covariance functions, there is a separate correlation length for each input parameter. This can be beneficial when the different input parameters have different influences on the output parameters (Rasmussen and Williams, 2006). In this work, the anisotropic covariance functions are named with the prefix ARD (automatic relevance determination).

For more information on Kriging, the reader is referred to Santner et al. (2018) and Rasmussen and Williams (2006).

The Kriging meta-model is used in this work to predict fatigue loads, i.e. short-term DELs $S_{eq}$, based on the input parameters $v_s$, TI, $H_s$, $T_p$, and $\theta_{mis}$. These environmental parameters were identified as significant in sensitivity analyses for operating wind turbines by Hübler et al. (2017a), Murcia

(2018), and Velarde et al. (2019). Due to their significant influence on the fatigue loads of operating wind turbines, it is assumed in this work that these parameters also have a significant impact on the fatigue loads in the case of an idling wind turbine.

It should be mentioned that a separate meta-model is created for each of the six internal forces analysed (see Table 1). To train the meta-models, the short-term DELs $S_{eq}$, determined with the aeroelastic simulation model $f$ depending on the input parameters, are used.

$$S_{eq} = f(v_s, \mathrm{TI}, H_s, T_p, \theta_{mis}) \tag{4}$$

A Kriging meta-model is a mathematical model which, in principle, can also make unphysical predictions, such as negative short-term DELs. To avoid the prediction of negative DELs, the logarithm of the short-term DELs is used to train the meta-models $g$. [TS2]

$$\ln(S_{eq}) = f(v_s, \mathrm{TI}, H_s, T_p, \theta_{mis}) \tag{5}$$

As the meta-models $g$ were trained with $\ln(S_{eq})$, the meta-models $g$ also predict values for $\ln(\hat{S}_{eq})$, where $\hat{S}_{eq}$ is the approximated short-term DEL.

$$\ln(\hat{S}_{eq}) = g(v_s, \mathrm{TI}, H_s, T_p, \theta_{mis}) \tag{6}$$

The values of Eq. (6) can then be used to calculate $\hat{S}_{eq}$ as follows.

$$\hat{S}_{eq} = \exp(\ln(\hat{S}_{eq})) \tag{7}$$

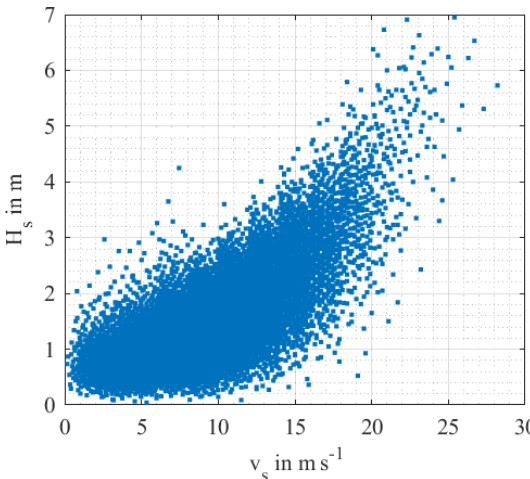

**Figure 4.** Correlations of mean wind speed $v_s$ and significant wave height $H_s$.

It should be noted that a Kriging meta-model generally predicts a mean and a corresponding standard deviation of the short-term DEL. For the prediction of the short-term DELs in this work, only the predicted mean values are used, as the mean values are the central values of the predictive distributions and provide a clear and unambiguous estimate of the expected short-term DELs.

## 3.2 Generation of training and test data

Using the previously defined initial transients, the training and test data sets for meta-modelling can be created. Here, the same simulation model and environmental conditions are used as described in Sect. 2.1. The five environmental input parameters for the meta-models, $v_s$, TI, $H_s$, $T_p$, and $\theta_{mis}$, are considered scattering parameters. In addition to these variables, $\psi$ and the random wind and wave seeds are also varied randomly in each simulation. For the other environmental variables, which are also scattered in reality (e.g. wind shear exponent), the mean values of their probability distributions are used.

In the sampling process, correlations between the input parameters are taken into account, as shown in Fig. 4. This means that combinations of environmental variables that do not occur in reality, such as a high wind speed in combination with a very low wave height, are not considered in this work. The correlations were considered using the statistical distributions of Hübler et al. (2017b), which contain the correlations between the different environmental variables.

A total of 10 000 samples are created using a Halton sequence. The Halton sequence is also known as a quasi-random sampling method. Here, samples are generated in the interval of [0, 1] with the help of an equation. The created samples are then transformed using the inverse cumulative probability distribution. The Halton sequence was selected

because it led to the best results in terms of the required number of samples compared to other methods in an earlier study (Müller et al., 2022) for an offshore wind turbine in operation. In addition, the Halton sequence was also used by Dimitrov et al. (2018) and Slot et al. (2020).

The resulting time series for the different internal forces (see Table 1) are subsequently transformed into short-term DELs using Eq. (1). The data set is then split into a training data set and a test data set. Here, 85 % of the samples are used to train the Kriging meta-model, and 15 % of the samples are then used to test the meta-model.

## 3.3 Load comparison of operating and idling wind turbines

As already described, idling conditions can certainly have a significant impact on the lifetime, as wave loads have a more significant impact on structural behaviour due to the almost-nonexistent aerodynamic damping. Thus, wave loads excite the structure more strongly. To understand the magnitude of the internal forces of the idling conditions compared to the internal forces of normal operation, the internal forces of the idling wind turbine and the internal forces of the operating wind turbine are compared first. This comparison also helps to check whether it is appropriate to create a meta-model for all the internal forces considered in this work or whether some of the internal forces in idling conditions are so small that no meta-model is required, as they have almost no influence on the lifetime.

For this investigation, the generated training and test data sets from the previous section are compared with the training and test data sets for meta-modelling for normal operation of the same offshore wind turbine (i.e. the same simulation model) from Müller et al. (2022). For both data sets, the wind speed range from $3\,\mathrm{m\,s^{-1}}$ (cut-in wind speed) to $25\,\mathrm{m\,s^{-1}}$ (cut-off wind speed) is divided into bins with a width of $2\,\mathrm{m\,s^{-1}}$. Subsequently, the mean of the short-term DELs is calculated for each bin and for both data sets. Example results are shown in Fig. 5 for $F_{x,\mathrm{DEL}}$, $F_{y,\mathrm{DEL}}$ (see Table 1), and the out-of-plane (OoP) bending moment at the blade root $M_{\mathrm{OoP,Root,DEL}}$. The comparison for the remaining internal forces is shown Appendix A in Fig. A4. It becomes clear that the internal forces at the monopile under idling conditions are in the same range compared to the internal forces during normal operating conditions and can be even higher than in normal operating conditions. The bending moments at the rotor blade roots are smaller but at least of a similar order of magnitude for low wind speeds. This study therefore leads to the conclusion that meta-models should be created for all six internal forces considered.

## 3.4 Input parameters for meta-modelling

Based on the results in Sect. 2.2 showing significant impacts of $\psi$ and the rotor speed on $M_{y,\mathrm{Root}}$, this section briefly in-

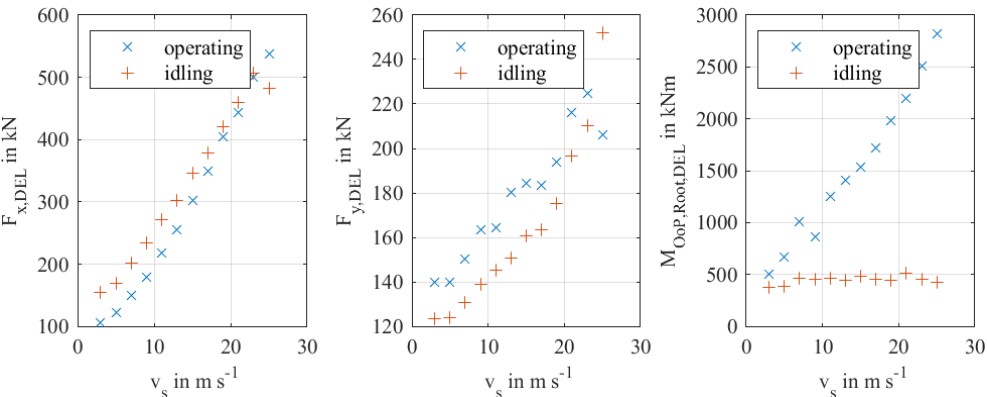

**Figure 5.** Comparison of the internal forces between idling and operating conditions for $F_{x,\text{DEL}}$, $F_{y,\text{DEL}}$, and $M_{\text{OoP,Root,DEL}}$.

vestigates whether $\psi$ and the mean rotor speed $\omega$ need to be regarded as additional input parameters for meta-modelling. For this purpose, the meta-models are first trained with the five environmental input parameters described in Sect. 3.2 ($v_s$, TI, $H_s$, $T_p$, and $\theta_{\text{mis}}$) using all 8500 training samples. Subsequently, meta-models are created with $\psi$ and $\omega$ as additional input parameters. To decide whether the additional input parameters need to be taken into account, the two meta-models created with different numbers of input parameters are compared for each considered internal force using the 1500 test samples. Here, the predicted short-term DELs of the two meta-models are compared with the short-term DELs determined with the aeroelastic simulation code FASTv8.

Example results are shown in Fig. 6 for $M_{y,\text{DEL}}$ and $M_{y,\text{Root,DEL}}$ depending on $v_s$. From Fig. 6a and c, it is clear that the use of the five input parameters for the approximation of $M_{y,\text{DEL}}$ leads to good results as both the overall trend and the scatter are represented. The meta-model of $M_{y,\text{Root,DEL}}$, in contrast, cannot represent the scatter of the simulated values of $M_{y,\text{Root,DEL}}$. Figure 6b and d show the approximations of the meta-models for $M_{y,\text{DEL}}$ and $M_{y,\text{Root,DEL}}$ with $\psi$ and $\omega$ as additional input parameters. It is clear that there is barely any change in the approximation of $M_{y,\text{DEL}}$ compared to the meta-model with five input parameters. For $M_{y,\text{Root,DEL}}$, in contrast, it becomes apparent that the approximation of the meta-model is significantly better when the two additional input parameters are employed. From these results it can be concluded, on the one hand, that it is sufficient for the internal forces at the monopile to use the five environmental input parameters that are also used for a wind turbine in normal operation. For the bending moments at the rotor blade roots, on the other hand, the two additional input parameters $\psi$ and $\omega$ are required for an acceptable approximation quality of the meta-models.

To train and test the meta-models, the parameters $\psi$ and $\omega$ are taken from the results of the aeroelastic simulations of the training and test data sets. Here, $\psi$ is the initial azimuth angle at the start of the 10 min period used to calculate the DEL. $\omega$

is the mean rotor speed during the entire 10 min period. For a later prediction with the meta-models, a value between 0 and 360° can be chosen arbitrarily, as the rotor position does not depend on any other parameter. However, $\omega$ depends on, among other factors, the wind field of the aeroelastic simulation and is therefore actually an output parameter of the simulation. In order to be able to use $\omega$ as an input parameter, the relationship between the input parameters, $v_s$, TI, $H_s$, $T_p$, $\theta_{\text{mis}}$, and $\psi$, and the output parameter, $\omega$, is therefore modelled using an additional Kriging meta-model. To train this meta-model for $\omega$, the same simulation data are used as for training the meta-models for the prediction of the short-term DELs. Since the rotor speed strongly depends on the random seeds (see also Sect. 2.2), it is not sufficient for the prediction of $\omega$ to only use the mean value of the Kriging meta-model for $\omega$ (as done for the meta-models for the prediction of the short-term DELs). Using only the mean value for the prediction, the scattering of $\omega$ cannot be properly modelled (see Fig. 7a). For this reason, the standard deviation is used in addition to the mean value to predict the short-term DELs. For this purpose, each predicted short-term DEL is randomly sampled from the normal distribution resulting from the predicted mean value of the corresponding short-term DEL and the standard deviation given by the Kriging meta-model of $\omega$. Including the standard deviation of the Kriging meta-model for $\omega$ allows for a sufficiently accurate mapping of the relationship between $\omega$ and the input parameters ($v_s$, TI, $H_s$, $T_p$, $\theta_{\text{mis}}$, and $\psi$). This is shown in Fig. 7b.

## 3.5 Investigation of Kriging settings and convergence study

Having defined the input parameters for meta-modelling, the next step is to analyse which meta-model settings lead to the best approximation of the simulation model. Here, four different basis functions and five covariance functions are analysed for the investigation of the meta-model settings for the Kriging meta-model. The analysed basis and covariance functions have already been used for Kriging meta-models

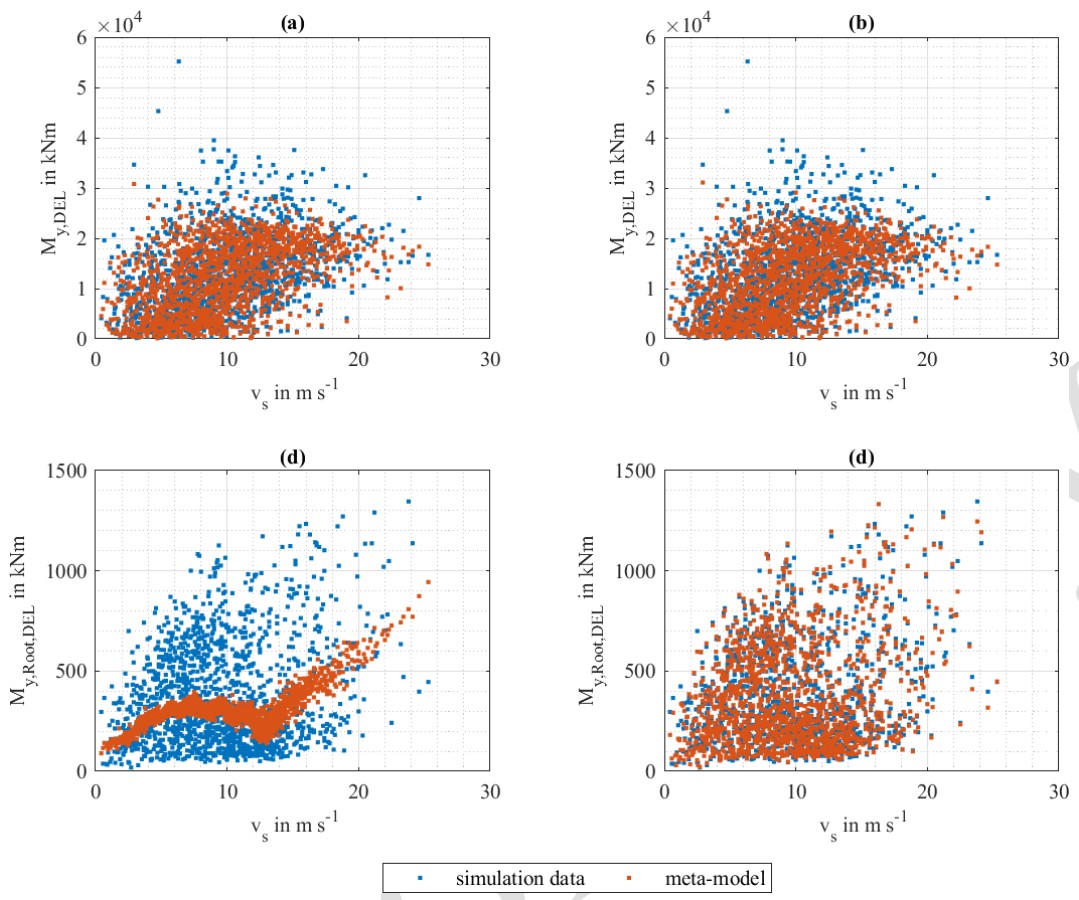

**Figure 6.** Comparison of two Kriging meta-models for $M_{y,\text{DEL}}$ and $M_{y,\text{Root,DEL}}$ with different numbers of input parameters: **(a)** and **(c)** use five input parameters ($v_s$, TI, $H_s$, $T_p$, $\theta_{\text{mis}}$), and **(b)** and **(d)** use seven input parameters, namely input parameters from **(a)** with $\psi$ and $\omega$ from FAST.

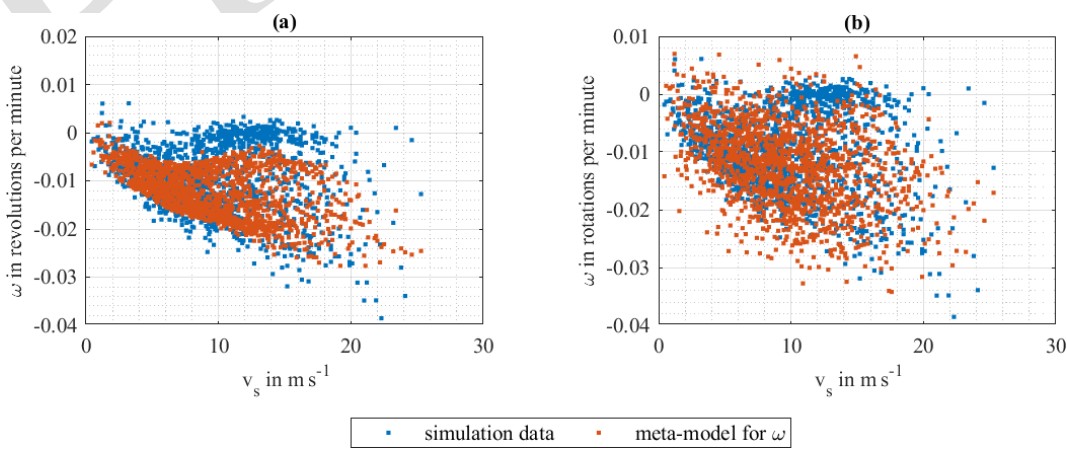

**Figure 7.** Approximation of $\omega$ by a Kriging model. **(a)** Prediction using the mean value of the Kriging model and **(b)** prediction using a random sampling of the normal distribution resulting from the mean values and the standard deviation predicted by the Kriging model.

**Table 3.** Investigated basis and covariance functions.

| Basis functions | none, constant, linear, and pure quadratic |
|---|---|
| Covariance functions | exponential, squared exponential, rational quadratic, matern 5/2, matern 3/2, ARD exponential, ARD squared exponential, ARD rational quadratic, ARD matern 5/2, ARD matern 3/2 |

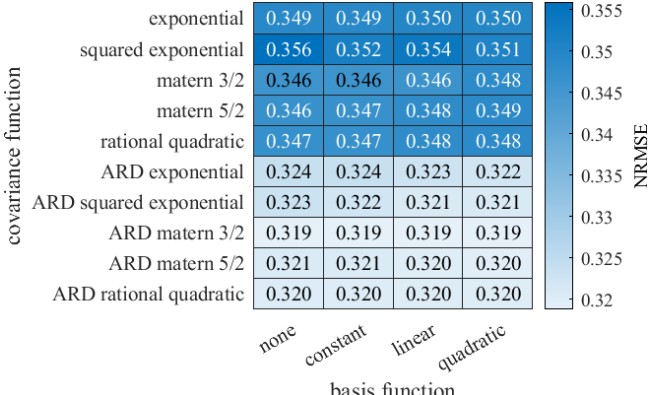

**Figure 8.** NRMSE for different settings of Kriging using $M_{y,\text{DEL}}$ as an example and using the entire training data of 8500 samples.

in wind energy. For the five covariance functions, both the option with equal correlation lengths (isotropic covariance functions) and the option with different correlation lengths (anisotropic covariance functions) for each input parameter are evaluated. The investigated basis and covariance functions are listed in Table 3. More detailed information on the covariance functions can be found in Rasmussen and Williams (2006).

For every possible combination of the investigated basis and covariance functions, a Kriging meta-model is created to select a meta-model setting for all further investigations in this paper. As a measure of error, the normalised root mean square error (NRMSE) is used.

$$\text{NRMSE} = \frac{1}{E(S_{\text{eq}}(\mathbf{x}))}\sqrt{\frac{\sum_{i=1}^{N_{\text{test}}}(\hat{S}_{\text{eq}}(\mathbf{x}_i) - S_{\text{eq}}(\mathbf{x}_i))^2}{N_{\text{test}}}}. \quad (8)$$

Here, $S_{\text{eq}}(\mathbf{x})$ is the simulated short-term DEL, and $\hat{S}_{\text{eq}}(\mathbf{x}_i)$ is the $i$th short-term DEL predicted by the meta-model. $N_{\text{test}}$ is the number of test samples, and $E(S_{\text{eq}}(\mathbf{x}))$ is the expected value of the short-term DELs $S_{\text{eq}}(\mathbf{x})$. One advantage of this error metric is the possibility to compare the meta-models for the different fatigue loads directly with each other. Therefore, it is quite simple to determine whether there are internal forces that can be better represented by a meta-model than others.

It turns out that the approximation quality mainly depends on whether an isotropic or anisotropic covariance function is chosen. The anisotropic covariance function, i.e. different correlation lengths for the different input parameters, leads to a better approximation of the simulation model than the isotropic covariance function. An example can be seen in Fig. 8 for $M_{y,\text{DEL}}$. This result was to be expected, as the five input parameters (fatigue loads at the monopile) or seven input parameters (fatigue loads at the rotor blade root) each have a different impact on the investigated internal forces.

To be able to decide which settings lead to the best meta-model – also with regard to the required number of samples – a convergence study is carried out. Here, the number of samples is increased step by step up to the maximum number of training samples of 8500. Due to the result that the approximation quality is better when using anisotropic covariance functions, only the anisotropic covariance functions are con-

sidered further for the convergence study. Figure 9 shows the convergence of the NRMSE for $M_{x,\text{Root,DEL}}$ and $M_{y,\text{Root,DEL}}$ for the five different anisotropic covariance functions in combination with all analysed basis functions as a function of the number of samples used. The results of the remaining internal forces are shown in Appendix A in Fig. A5. It becomes clear that the choice of the covariance function has an influence on the number of samples required for meta-modelling, while the influence of the choice of the basis function is less significant for the most internal forces.

Since the ARD matern 3/2 covariance function leads to the best results overall for all fatigue loads considered, this covariance function is used from this point onwards in this paper. For the basis function, the use of no basis function, a constant basis function, or a linear basis function is recommended, as in combination with the ARD matern 3/2 covariance function, these lead to a good approximation for all internal forces analysed. Therefore, in the following, a linear basis function is used.

Furthermore, it is clear from Figs. 9 and A5 that the value of the NRMSE has not yet converged for all internal forces (e.g. $M_{x,\text{Root,DEL}}$ and $M_{x,\text{DEL}}$), even when all 8500 training samples are used. In order to achieve convergence, further samples would be necessary. However, as this is only useful if a convergence of the NRMSE is required for a sufficiently accurate meta-model, the next section first analyses how suitable the meta-models created in this work, using all 8500 samples, are in comparison to the simulation model.

## 3.6 Evaluation of the approximation quality of the Kriging meta-model

In this section, the previously raised question of how good the meta-models created are in terms of their approximation quality and whether they should or could be more accurate is addressed. Here, it should be noted that the aeroelastic simulations themselves also contain uncertainties, e.g. due to the random seeds used for the generation of the wind

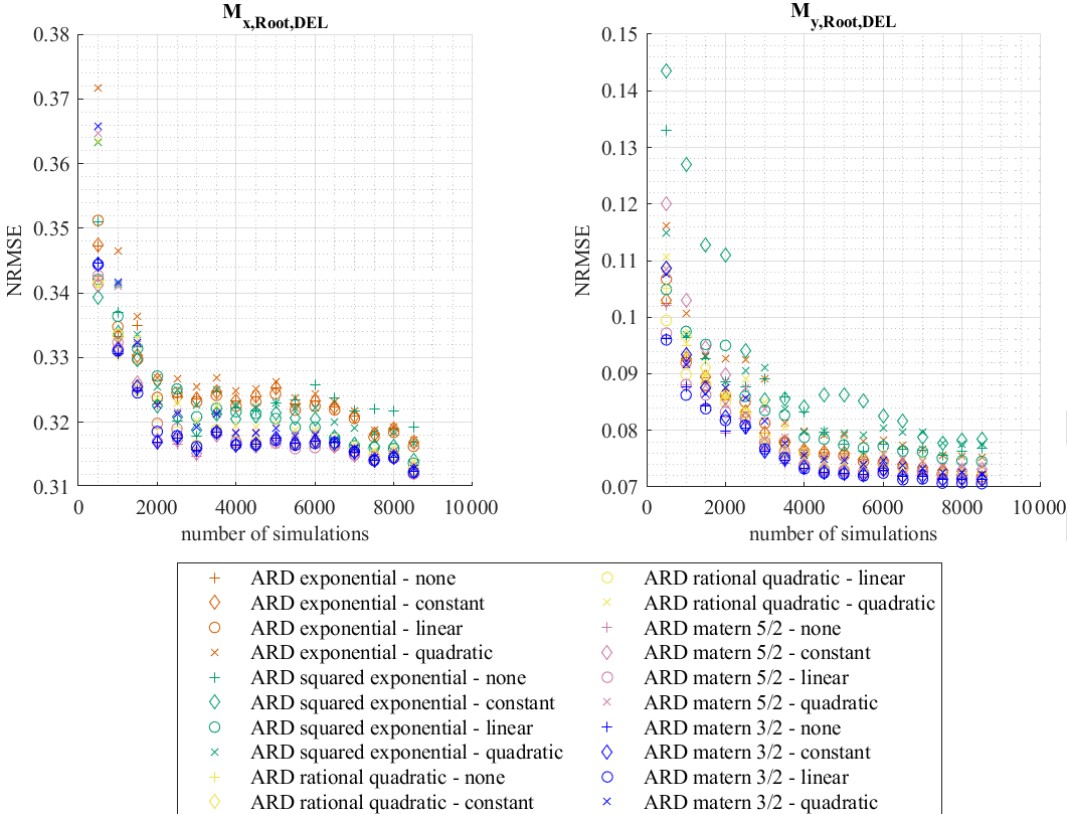

**Figure 9.** NRMSE depending on the number of simulations for different settings of the Kriging meta-model for the bending moments at the rotor blade root. For the prediction of the bending moments at the rotor blade root, the values of $\psi$ and $\omega$ were taken from the results of the aeroelastic simulations of the test data set to ensure that the predicted short-term DELs and the short-term DELs of the test data set are comparable.

and wave fields. This seed-to-seed uncertainty and the uncertainty in the meta-models compared to the aeroelastic simulation model are analysed in the following. The aim is to be able to rate the approximation quality of the meta-model in comparison to the seed-to-seed uncertainty of the simulation model. For this purpose, the test data set (see Sect. 3.2) is simulated nine additional times with varied wind and wave seeds using the aeroelastic simulation code FASTv8. For the resulting 10 test data sets, which are identical except for the wind and wave seeds, the NRMSE is calculated for all possible combinations of the 10 test data sets to determine the seed-to-seed uncertainty of the simulation model (NRMSE$_{sim}$, 45 combinations in total). A schematic overview of the combinations considered can be found in Table 4. The mean value NRMSE$_{sim,mean}$ is then calculated from the 45 values of NRMSE$_{sim}$. At the same time, the predictions of the meta-models are compared with the simulated test data sets. For the meta-models of the rotor blades, the predictions are compared with the test data sets in the same way that the test data sets were previously compared with each other (see Table 4). Here, only the "S" on the right-hand side is replaced by an "M" for meta-model prediction.

For M2 to M10, separate meta-model predictions are generated. The number of the meta-model prediction, e.g. M2, indicates that the values of $\psi$ from test data set 2 (S2) are used for this prediction. For $\omega$, the rotor speeds are predicted using the meta-model for $\omega$ with the values of $\psi$ from S2. Unlike the rotor blades, there is no dependence on $\psi$ and $\omega$ of the meta-models for the monopile. For this reason, there is only one prediction from the meta-model for the test data for each internal force. This reduces the number of combinations with the test data sets to a total of 10 possible combinations (S1 vs. M1 to S10 vs. M10). NRMSE$_{meta}$ is calculated for each combination of meta-model prediction and test data set. Then, in the next step, the mean values NRMSE$_{meta,mean}$ are determined from the 45 (rotor blades) and 10 (monopile) values of NRMSE$_{meta}$. The resulting mean values of the NRMSE are then compared for all considered internal forces (see Table 5). Additionally, a visual comparison, in which the first test data set (S1) is plotted against the second test data set (S2) and against the predictions of the meta-model (M2), is conducted. An example is shown in Fig. 10 for $F_{x,DEL}$ and $M_{y,Root,DEL}$.

**Table 4.** Schematic overview for determining the seed-to-seed uncertainty in the different test data sets with different random seeds. For example, the simulated short-term DELs of test data set 1 (S1) are compared with the simulated short-term DELs of test data sets from S2 to S10, and the NRMSE is then calculated for each combination.

| S1 | S2 | S3 | S4 | S5 | S6 | S7 | S8 | S9 | S10 |
|----|----|----|----|----|----|----|----|----|-----|
| S2 | S3 | S4 | S5 | S6 | S7 | S8 | S9 | S10 | |
| S3 | S4 | S5 | S6 | S7 | S8 | S9 | S10 | | |
| S4 | S5 | S6 | S7 | S8 | S9 | S10 | | | |
| S5 | S6 | S7 | S8 | S9 | S10 | | | | |
| S6 | S7 | S8 | S9 | S10 | | | | | |
| S7 | S8 | S9 | S10 | | | | | | |
| S8 | S9 | S10 | | | | | | | |
| S9 | S10 | | | | | | | | |

**Table 5.** Comparison of the mean values of $NRMSE_{sim}$ for simulations against simulations resulting from all possible combinations of 10 random seeds (45 combinations) with the mean values of $NRMSE_{meta}$ for the meta-model against simulations resulting from the possible combinations of the same 10 random seeds (rotor blades: 45 combinations, monopile: 10 combinations) for all considered internal forces using the maximum number of training samples. $NRMSE_{meta,mean}$ values are calculated using $\omega$ from the meta-model of $\omega$.

| Internal force | $NRMSE_{sim,mean}$ | $NRMSE_{meta,mean}$ |
|---|---|---|
| $F_{x,DEL}$ | 0.198 | 0.146 |
| $M_{y,DEL}$ | 0.439 | 0.318 |
| $F_{y,DEL}$ | 0.149 | 0.114 |
| $M_{x,DEL}$ | 0.355 | 0.263 |
| $M_{x,Root,DEL}$ | 0.475 | 0.374 |
| $M_{y,Root,DEL}$ | 0.681 | 0.697 |

The results in Table 5 show that, with the exception of $M_{y,Root,DEL}$, $NRMSE_{meta,mean}$ is comparable to $NRMSE_{sim,mean}$ for all considered internal forces. However, the deviation for $M_{y,Root,DEL}$ is very small at less than 3 %, meaning that the NRMSE values are similar. For this reason, it can be concluded that the meta-models are sufficiently accurate, as $NRMSE_{meta,mean}$ is comparable to or smaller than $NRMSE_{sim,mean}$. This is also clear from Fig. 10.

Based on these results and an additional convergence study, where the determination of $NRMSE_{meta,mean}$ was repeated for training sample sizes from 500 to 8500 (see Fig. A6), the findings of the previous section regarding the required sample size and the choice of the meta-model settings must be re-evaluated. Assuming that the values of $NRMSE_{meta}$ do not have to be smaller than the values for $NRMSE_{sim}$, the results show that significantly fewer simulations than the 8500 simulations carried out in this work are enough to create sufficiently good meta-models. When using a linear basis function and an ARD matern 3/2 covariance function as recommended in the previous section,

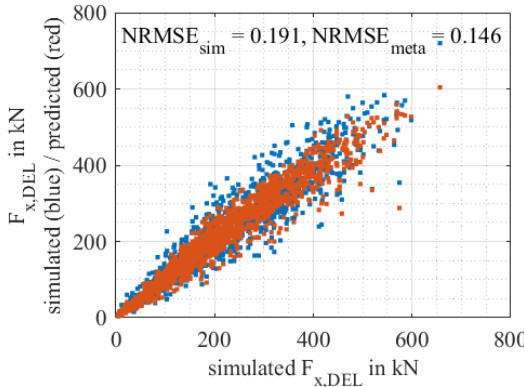

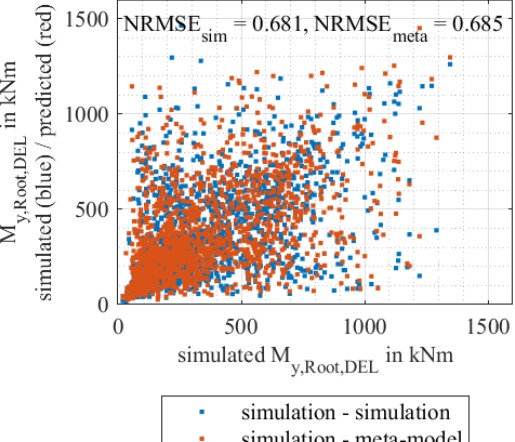

**Figure 10.** Comparison of the seed-to-seed uncertainty in test data sets S1 and S2 (blue data points) with the predictions of the meta-models (red data points) for $F_{x,DEL}$ and $M_{y,Root,DEL}$. For the prediction of $M_{y,Root,DEL}$, $\psi$ is taken from test data set S2, and $\omega$ is predicted with the meta-model of $\omega$.

as few as 2500 samples are sufficient to create the meta-models for every considered internal force, although the NRMSE (see Figs. 9 and A5) has not yet converged. Here, the $NRMSE_{meta,mean}$ has almost converged for all internal forces (deviation of $NRMSE_{meta,mean}$ using 8500 samples is less than 5 %). Of course, the choice of the sample size depends on the requested accuracy of the meta-models. Even with 500 samples, a very good approximation can be achieved. It can therefore be summarised that significantly fewer samples are required to create a sufficiently accurate meta-model compared to the results of the convergence study in Sect. 3.5.

Figure 10 also clearly shows that the seed-to-seed uncertainty has a different impact on the considered internal forces. For $M_{y,Root,DEL}$ in particular, the influence of the seed-to-seed uncertainty is significant and it becomes clear that a change of the random seeds alone (while keeping all other input parameters unchanged) results in a large scatter-

ing of the short-term DELs. This large scattering can be explained by the fact that the different random seeds result in different wind fields, which cause the rotor to rotate at different speeds during the simulation (see also Sect. 2.2).

## 4 Comparison to meta-modelling of an operating wind turbine

The findings concerning the meta-modelling of the idling offshore wind turbine are compared with the findings concerning the meta-modelling of the same offshore wind turbine in normal operation. For this purpose, the results from a previous study are used, in which the meta-modelling of the same simulation model, i.e. the NREL 5 MW reference turbine on the OC3 monopile, was investigated (Müller et al., 2021). In this study, the same five input parameters were used for meta-modelling, and the use of the ARD matern 3/2 covariance function in combination with a quadratic basis function was recommended.

It is noticeable that the results of the investigations into meta-modelling of the short-term DELs are very similar for idling and for normal operation. The findings that an anisotropic rather than isotropic covariance function leads to a better meta-model and the influence of the basis function is rather low are valid for both normal operating and idling conditions. Only the recommendation of the use of no basis function, a constant basis function, or a linear basis function instead of a pure quadratic basis function differs. However, the results in Müller et al. (2021) show that it is also possible to use the recommended basis functions of this work for normal operation. It would therefore be possible to use the same covariance and basis functions for both operating states.

The five input parameters that were used for normal operation are also sufficient for meta-modelling of the idling condition for the prediction of the internal forces at the monopile. With regard to the approximation of the bending moments at the rotor blade roots, two additional parameters ($\psi$ and $\omega$) must be added for idling compared to normal operation. However, this does not affect the choice of the Kriging settings so that the same settings could be used for the Kriging meta-models for the prediction of the short-term DELs both at the monopile and at the rotor blade root.

With regard to the evaluation of the approximation quality, no conclusion can currently be made on the basis of the work in Müller et al. (2022) TS3 as the influence of the seed-to-seed uncertainty was not investigated in this study. Since the same simulation model was used, it can be assumed that similar effects occur in normal operation as during idling. Nevertheless, this would have to be confirmed in a further study.

## 5 Conclusions

In this work, meta-modelling for the estimation of damage equivalent loads for an idling offshore wind turbine was investigated for the first time. The simulation model analysed was the NREL 5 MW reference turbine on the OC3 monopile. Here, the approximation of the fatigue loads at the monopile at the mud line in the wind direction and perpendicular to the wind direction, as well as the blade root moments, was investigated. Moreover, before the meta-models were created, the run-in times for the idling wind turbine were determined. In this context, the following new findings have been achieved:

1. The investigation of the run-in times showed that the required run-in times decrease with increasing wind speed. It turned out that the recommended run-in times are between 60 and 800 s depending on the wind speed.

2. The findings concerning meta-modelling of the idling wind turbine are generally similar to the findings concerning meta-modelling of the same wind turbine in normal operation.

3. However, it was found that $\psi$ and $\omega$ have a significant impact on the flapwise bending moment at the rotor blade root $M_{y,\mathrm{Root}}$. Therefore, these parameters should be included in meta-models for the internal forces at the blade root. Nevertheless, there is no need to include them in meta-models for the internal forces at the monopile.

4. Regarding the meta-model settings, the use of the ARD matern 3/2 covariance function in combination with no basis function, a constant basis function, or a linear basis function is recommended for the creation of the meta-models.

5. Due to large seed-to-seed uncertainty in aeroelastic simulations, significantly fewer samples compared to the entire training data set of 8500 samples are required to create the meta-models (i.e. approximately 2500).

Nevertheless, there are some limitations to the generalisation of this work. With regard to the aeroelastic simulations in the time domain, only the run-in times were considered in this study. However, due to the slow change of $M_{y,\mathrm{Root}}$ caused by the slow rotation of the rotor, it is possible that not only the run-in time but also the total simulation time might be factors that should be investigated in future work. Here, the simulation time could be selected, for example, to be long enough to include one rotation of the rotor during the simulation time. In this case, the variation in the initial azimuth angle might no longer be as relevant for determining the DELs. Regarding the run-in times, it is important to keep in mind that the run-in times only apply to the simulation model of the NREL 5 MW reference turbine on the OC3 monopile analysed in this work using the aeroelastic simulation code FASTv8. If a different wind turbine, substructure, or aeroelastic simulation code is used, the run-in times may differ from those in this paper. Nevertheless, the run-in times given can, of course,

provide an indication of the choice of the run-in times. Furthermore, it should be noted that the meta-modelling was investigated only for six different internal forces at two locations of the offshore wind turbine and for one specific wind turbine model. To be able to assess whether these results can be transferred to other locations of the wind turbine or to other wind turbine models and substructures, further investigations must be carried out. It should also be noted that the meta-models were not created or tested for extreme environmental conditions. The meta-models were created for the input parameter combinations covered by the 8500 simulations of the training data and tested for the input parameter combinations of the 1500 test samples. In addition to the points discussed, it should be mentioned that the use of short-term DELs for the fatigue calculation, especially for the rotor blades, is a simplification.

In future work, the new meta-models created in this work can be used for the lifetime reassessment of wind turbines to enable the consideration of idling in the lifetime reassessment in addition to modelling normal operation using meta-models. This would lead to a significant reduction in the computational effort of the lifetime reassessment. However, even though the meta-models in this study show a good approximation quality, their use for lifetime reassessment should be investigated in a further study. Furthermore, the findings from this work can be used to create meta-models for other simulation models of idling wind turbines. In addition, other meta-models could also be investigated, for example, the use of a heteroscedastic Gaussian process regression. In contrast to the homoscedastic Gaussian process regression used in this work, heteroscedastic Gaussian process regression can take varying uncertainty (standard deviation) over the range of input parameters into account.

## Appendix A: Additional figures

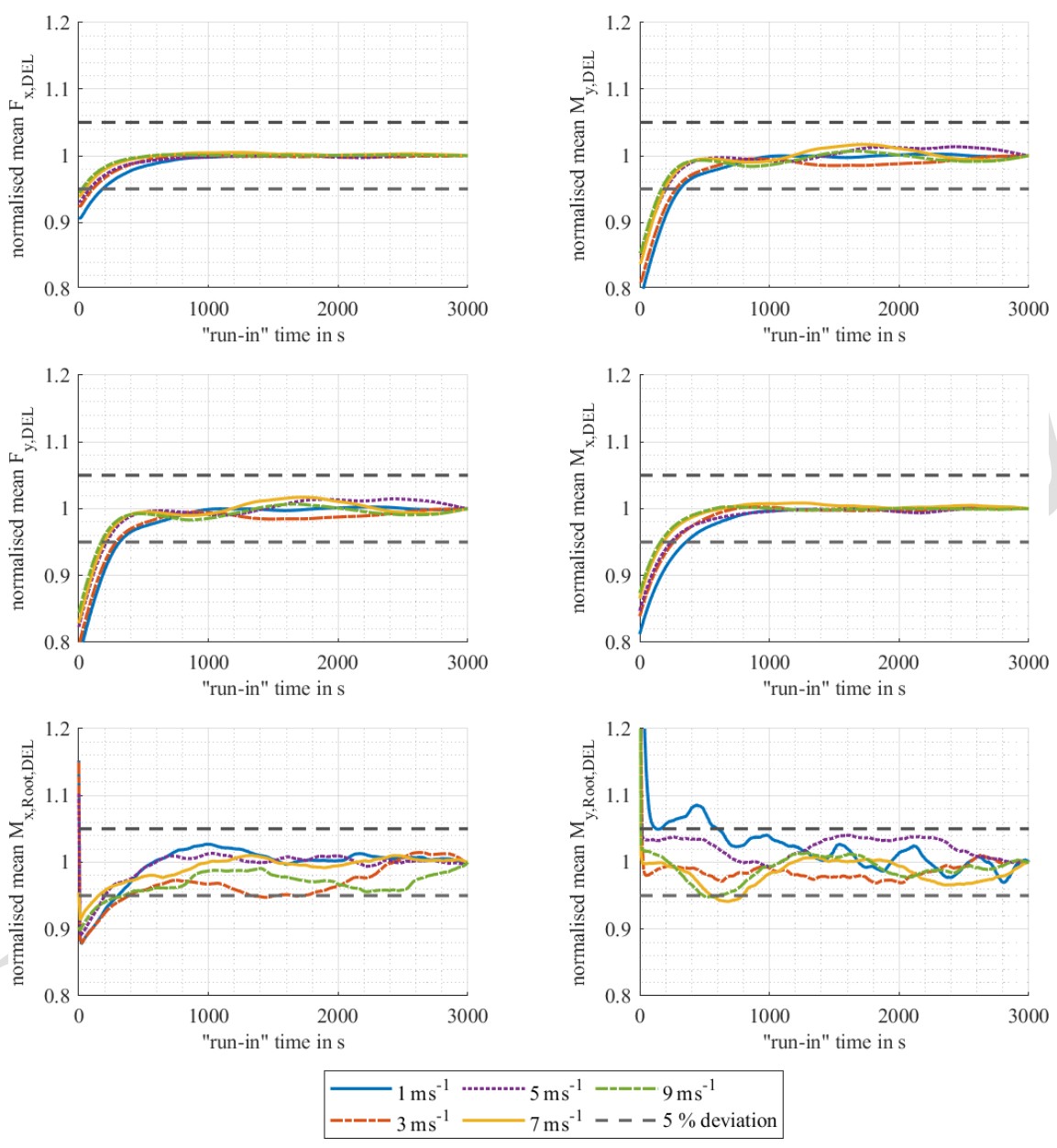

**Figure A1.** Initial transient behaviour of the idling wind turbine for mean wind speeds of 1 to $9\,\mathrm{m\,s^{-1}}$ for all considered internal forces.

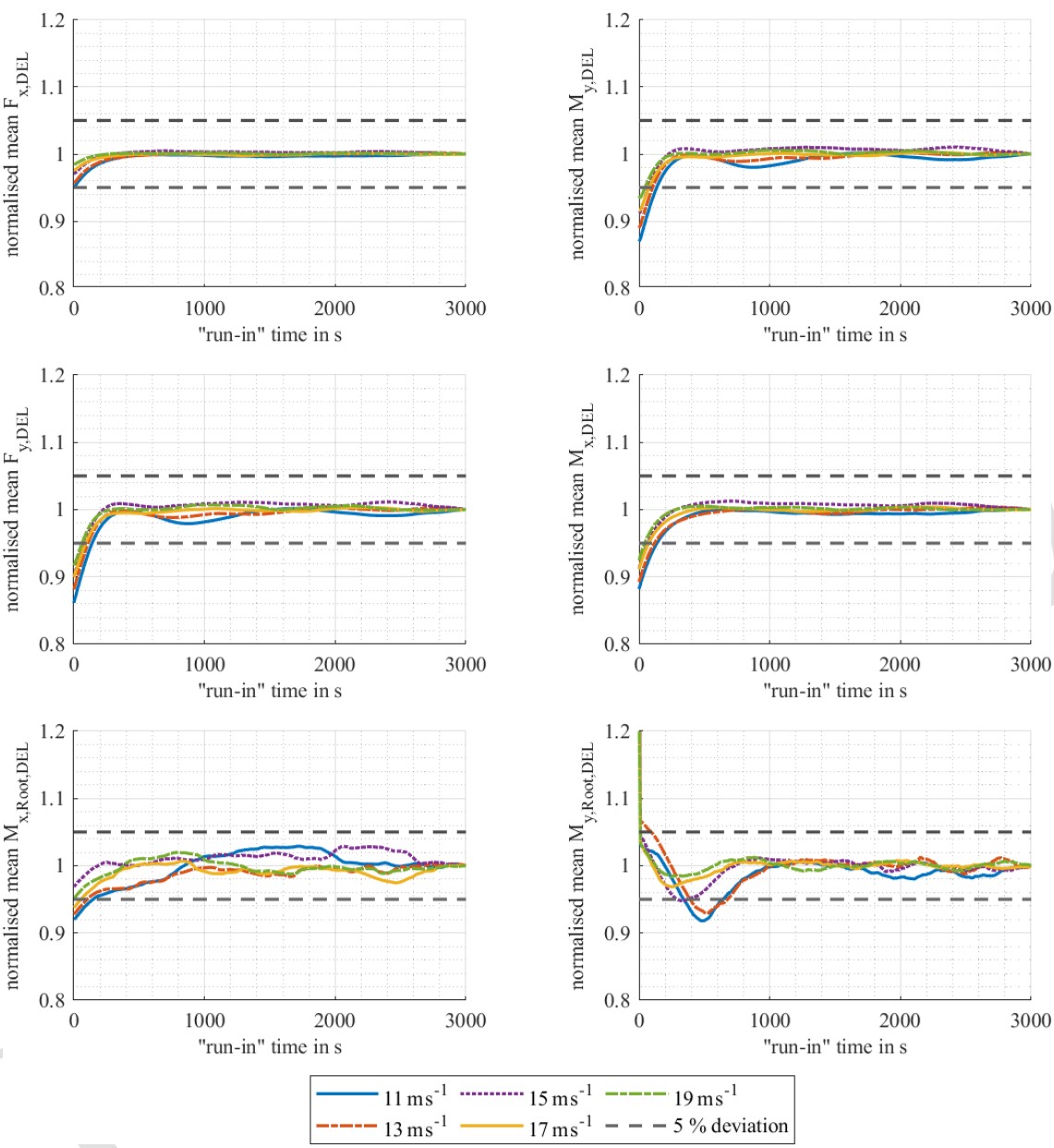

**Figure A2.** Initial transient behaviour of the idling wind turbine for mean wind speeds of 11 to 19 m s$^{-1}$ for all considered internal forces.

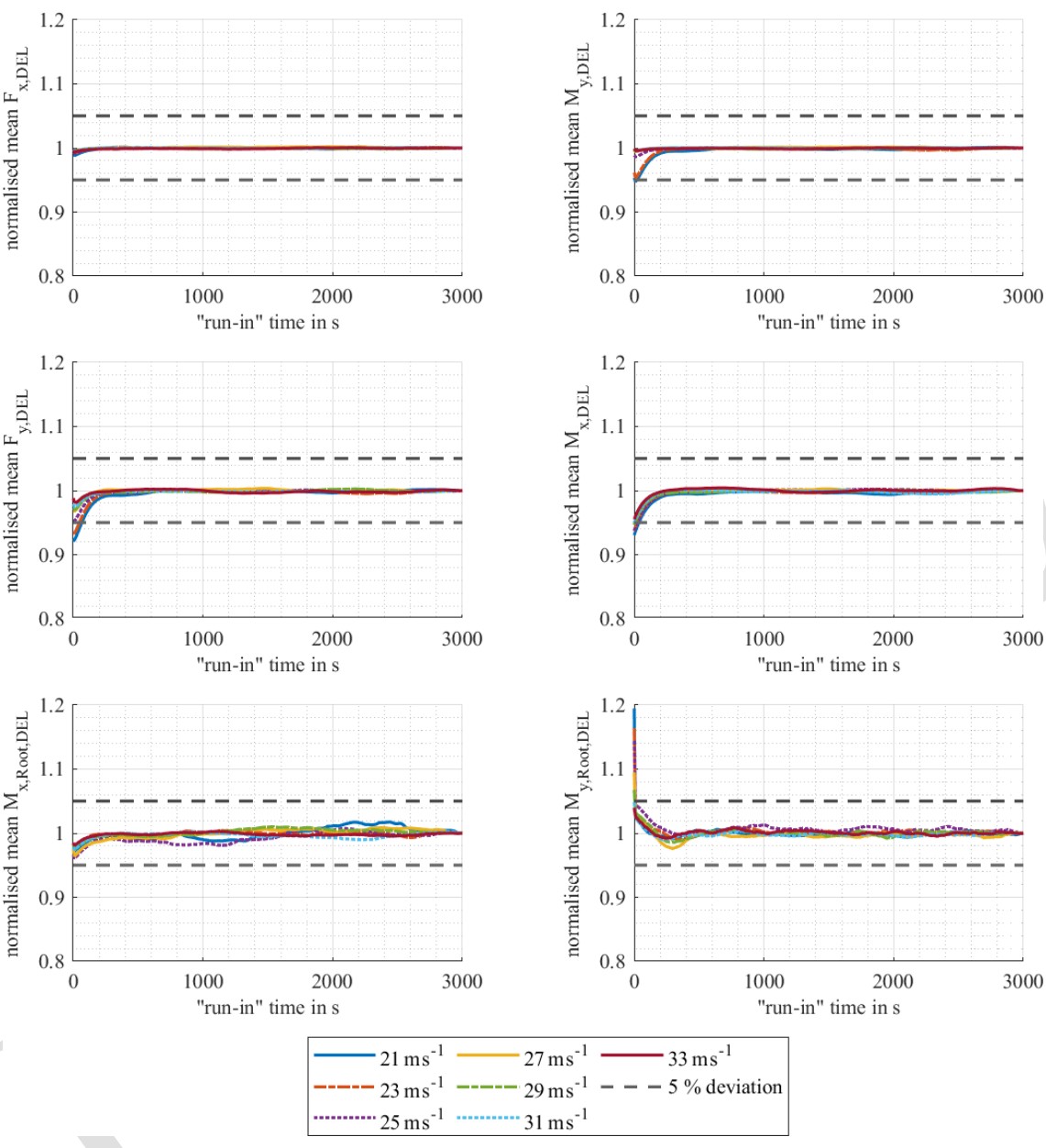

**Figure A3.** Initial transient behaviour of the idling wind turbine for mean wind speeds of 21 to 33 m s$^{-1}$ for all considered internal forces.

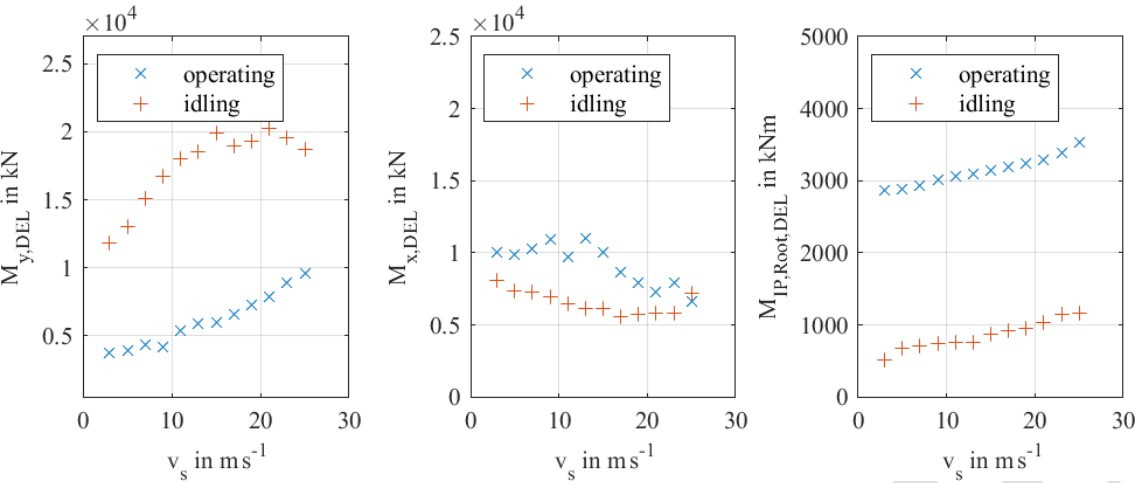

**Figure A4.** Comparison of the internal forces between idling and operating conditions for $M_{y,\mathrm{DEL}}$, $M_{x,\mathrm{DEL}}$, and the in-plane bending moment at the blade root $M_{\mathrm{IP,Root,DEL}}$.

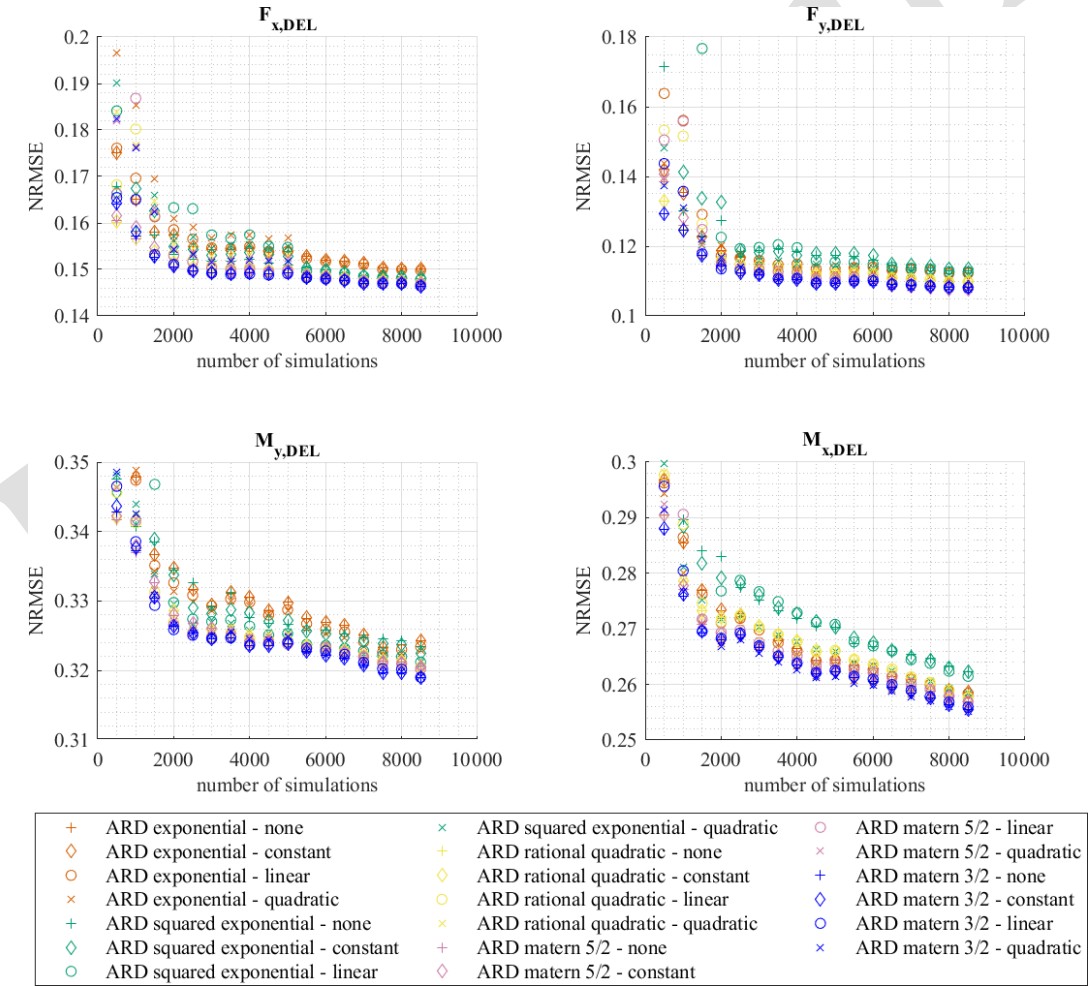

**Figure A5.** NRMSE depending on the number of simulations for different settings of the Kriging meta-model for the internal forces at the monopile.

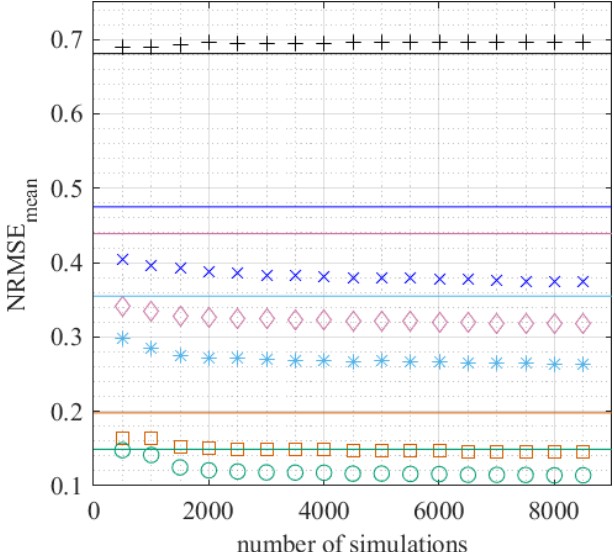

**Figure A6.** Convergence study of $NRMSE_{meta,mean}$ depending on the number of training samples. The solid lines are the corresponding values of $NRMSE_{sim,meta}$ for each internal force.

**Disclaimer.** Publisher's note: Copernicus Publications remains neutral with regard to jurisdictional claims made in the text, published maps, institutional affiliations, or any other geographical representation in this paper. While Copernicus Publications makes every effort to include appropriate place names, the final responsibility lies with the authors. Views expressed in the text are those of the authors and do not necessarily reflect the views of the publisher.

**Acknowledgements.** This work was supported by the LUH computing cluster, which is funded by Leibniz University Hannover, the Lower Saxony Ministry of Science and Culture (MWK), and the German Research Association (DFG).

**Financial support.** This research has been supported by the Bundesministerium für Wirtschaft und Energie (grant no. 03EE3029A).

The publication of this article was funded by the open-access fund of Leibniz Universität Hannover.

**Review statement.** This paper was edited by Weifei Hu and reviewed by two anonymous referees.

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

## Remarks from the typesetter