# Peer review of "Kriging meta-models for damage equivalent load assessment of idling offshore wind turbines"

_Wind Energy Science, 2025_

## Referee Comment (RC2)

*Discussion of:*

**Kriging meta-models for damage equivalent load assessment of idling offshore wind turbines**

06 September 2025

The manuscript presents a methodology for building surrogate models (metamodels) for predicting loads under idling conditions, using environmental conditions as inputs. The manuscript is well written and easy to follow. Below, some suggestions for improvement:

**General comments**

1) Novelty: I suggest some more emphasis on the concrete novel elements. Surrogate models are known, including Kriging, and many publications report a few thousand data points as sufficient for model training – and that one shouldn't necessarily use much more as the computational time increases significantly (especially for the Kriging model used by the authors). The novelty must be in studying how to approach the idling problem and if it has differences with the standard "normal operation" modelling approach.

2) I think the authors are mixing together two problems that affect convergence of the time series: the need for the numerical simulator to run-in, and the fact that the time series apparently have non-ergodic properties (impossible to properly characterize statistically) due to the very slow rate of change of some signals. In my view, the significant effort of the authors (tens of thousands of simulations of 60min duration) has in fact mostly been spent on dealing with the non-ergodic properties of the signal, which is a valid problem but unrelated to the run-in time. In fact, the total simulation time might be another factor to study. I suggest the authors to go deeper into this problem as this is a unique challenge for idling conditions.

3) I wonder what is the total contribution from idling events to the lifetime fatigue damage accumulated on a typical site? This could give a very tangible insight into the importance of doing this type of study, and I believe it should be simple for the authors to estimate the number.

**Specific comments**

4) Abstract: I think the first few sentences in the abstract are unclear. I wouldn't call the surrogate model an alternative to aeroelastic simulations, since one needs to run aeroelastic simulations to train the model. The surrogate model is literally a surrogate or a substitution of the aeroelastic simulations when repeated runs are needed – in that sense, the surrogate models help us by making sure we only need to run the high-fidelity simulation experiment once.

5) Page 2, line 39: IEC 61400-1, ed4 is the current standard. It also considers idling, so I would cite the most recent standard.

6) Page 2, lines 48-50: I think the use of "load cases" here is confusing. I believe the authors mean that approximately 5% of the operational lifetime (5% of the reference 10-minute periods) are spent

under idling conditions. In IEC61400-1 lingo however, "load cases" are specific scenarios (Design Load Cases, DLCs) to be simulated, and they don't necessarily correspond to a specific time fraction. The "idling" DLC is one – DLC6.4. I suggest replacing "load cases" with "10-minute realizations" or similar.

7) Page 2, lines 54-55: while 5-15% of the lifetime indeed corresponds to more than 100,000 ten-minute periods, I wouldn't agree that we need 100,000 simulations to cover it as normally the load cases will create a lookup table for a few combinations of wind conditions and apply probability weighting. For example, the normal production load case DLC1.2 normally requires anything between 200 and 1000 simulations, depending on the turbine technology and onshore/offshore configuration. I don't see why we would need more for idling.

8) Page 4, line 121-123: 10,000 simulations per wind speed to determine run-in times sounds excessive. With 16 wind speed bins, does this mean 160,000 simulations in total? And I believe the 2m/s step is not ideal if we are interested in the idling range, because we have most of the idling happening at very low wind speeds, and we may want higher resolution at those ranges.

9) Section 3.2, line 238: another 10,000 simulations are used to train the Kriging model. Maybe the simulations done for determining the run-in time can be reused instead?

---

## Author Comment (AC1)

**Kriging meta-models for damage equivalent load assessment of idling offshore wind turbines - Response to reviewers' comments:**

We want to thank both reviewers for their time and effort in reviewing our article and their constructive feedback that helped us to improve the quality of our paper.

In the following, you can find our answers to the comments. Here, the line numbers in the reviewers' comments refer to the first preprint version (initial submission) whereas the line numbers in our answers refer to the revised version of the paper. The edits in the manuscript are highlighted in magenta for reviewer 1 and in blue for reviewer 2. Adjustments made based on the comments from both reviewers are marked in cyan.

For your information, in relation to comment 2 from reviewer 2, we made some adjustments in Section 3.4 to 3.6 as we noticed a small mix-up regarding the initial rotor position as input parameter for the rotor blade meta-models. Instead of the initial rotor position (azimuth angle) of the 10-minute period used to calculate the DELs, we used the initial rotor position at the beginning of the aeroelastic simulation (i.e., before the run-in time) as input parameters for the meta-models. We corrected this during the revision, which has improved the meta-models. In the course of this, we revised the calculation method regarding the seed-to-seed uncertainty in Section 3.6. Based on these adjustments, the recommendations for the meta-model settings and the required training sample size were updated.

**Reviewer 1:** (marked magenta in the manuscript)**

1. Abstract: I would have appreciated a slightly less technical abstract, where the reader gets a higher level overview of what your article is about. Also, some terms generated some confusion even in a technical expert like me. I later understood what they refer to, but I had to google to confirm what "lifetime reassessments" and "run-in times" referred to. I was more familiar with wording like "lifetime extension potential" and "initial transients". I would strongly recommend clearing these possible sources of confusion, especially from the abstract that will hopefully be read by many people.

Thank you for your comment. We adjusted the abstract to make it less technical. We deleted the first sentence of the abstract, as although it explains the motivation for creating the meta-model, it is not particularly relevant to the general overview of the paper's content. We also reorded the second sentence (also based on the comments from reviewer 2). Now, in the abstract, we only refer to lifetime calculations which will fortunately make the abstract easier to understand. Furthermore, we removed the sentence from the abstract where we mentioned the initial transients to make the abstract less technical and prevent confusion at this point.

In addition to the adjustments in the abstract we also made minor changes to the introduction to clarify the terms "lifetime reassessment" and "run-in times". We added a sentence right at the beginning of the introduction (see lines 16 to 17) to make it clear that a lifetime recalculation or lifetime reassessment is required to determine the lifetime extension potential. For the "run-in times" respectively the "initial transients", we also slightly adjusted the description (see lines 78 to 81) to make it clear that we are referring to the initial transients and that by run-in times we mean the time period of the initial transients. For the remaining parts of the paper, we decided to retain the

term "run-in times" as this is besides "initial transients" also a common term used to describe the time period for the initial transients.

2. Line 20: The load cases are lumped here. "here" where?

You're right. This is a bit confusing. The load cases are lumped in the two studies we mentioned before. We changed the sentence a little bit to make it clearer (see line 24).

3. Line 42: lower occurring loads. Fatigue or peak loads?

Here, with "lower occurring loads" the fatigue loads are meant, as we are referring to the effects on the lifetime. Peak loads are another topic, but we assume that these are also lower compared to normal operation.

4. Lines 42-47: a citation seems missing

You are right, citations are missing here. We added three references. The first study of Ziegler et al. (2024) shows that idling leads to lower loads for onshore wind turbines and small wind turbines on monopiles compared to normal operation. In Santos et al. (2024) the fatigue loads of a standard monopile and a XL monopile considering normal operation and idling are compared. Since small (standard) monopiles were included in both references, we extended the statement that during idling operation, the loads on onshore wind turbines are lower than during normal operation to small offshore wind turbines on monopiles (see line 47). The third study, we added is a study of Velarde et al. (2020). In this study, Velarde et al. show for a 10 MW offshore wind turbine on a monopile that especially for large wind turbines on monopiles, the impact of the wave loads is large. The references can be found in lines 48 to 53.

5. Line 49: "of of", typo

Thank you for pointing out the word repetition here. We corrected that.

6. Line 65: "known" typo

Thank you. We corrected that.

7. Line 71: I am more familiar with the wording "initial transients" rather than "run-in times"

Thank you for your comment. Please see our response to point 1.

8. Line 95: There is no action item here, but please note that FASTv8 is 10+ years old. I'd strongly recommend upgrading to the latest releases of OpenFAST to make your work more impactful

That's right, FASTv8 is already quite old. For this reason, we have already started switching to OpenFAST for other types of wind turbines in a recent project. However, since the investigations in this paper should be consistent with our preliminary work (see Müller et al. (2021), Müller et al. (2022) and Schmidt et al. (2023)), we decided to continue using the "old" FASTv8 version.

9. Line 255: I'd recommend referencing Table 1 here, it took me a while to understand what those forces and moments were

Thank you for pointing that out. We added a reference to Table 1 in this line (line 267). We hope that makes it easier to identify the forces and moments.

**Reviewer 2:** (marked blue in the manuscript)**

**General comments**

1) Novelty: I suggest some more emphasis on the concrete novel elements. Surrogate models are known, including Kriging, and many publications report a few thousand data points as sufficient for model training — and that one shouldn't necessarily use much more as the computational time increases significantly (especially for the Kriging model used by the authors). The novelty must be in studying how to approach the idling problem and if it has differences with the standard "normal operation" modelling approach.

Thank you very much for this important comment. We modified the abstract to better highlight the novelty of this work (lines 5 to 9). It is now being emphasised more clearly that this is the first time a meta-model has been created for an idling offshore wind turbine and that the findings made in studies on turbines in normal operation cannot simply be transferred directly. We removed the sentence about the "run-in" times in the abstract and hope that this will better highlight the result that two additional input parameters must be taken into account for the meta-models of the rotor blades. With these adjustments, we hope that the novelty of our work will now be better appreciated. Furthermore, in the introduction in lines 71 to 72, we also reworded the sentence a little bit to make it clearer that this is the first time a meta-model is being investigated in detail (it has already been described above that the findings from normal operation cannot simply be transferred to idling).

In the conclusion in line 415, we also added a few words to make it clearer that this is the first time that a meta-model for an idling offshore wind turbine has been investigated.

2) I think the authors are mixing together two problems that affect convergence of the time series: the need for the numerical simulator to run-in, and the fact that the time series apparently have non-ergodic properties (impossible to properly characterize statistically) due to the very slow rate of change of some signals. In my view, the significant effort of the authors (tens of thousands of simulations of 60min duration) has in fact mostly been spent on dealing with the non-ergodic properties of the signal, which is a valid problem but unrelated to the run-in time. In fact, the total simulation time might be another factor to study. I suggest the authors to go deeper into this problem as this is a unique challenge for idling conditions.

Thank you for your comment. You are right, we spent a significant effort on dealing with the non-ergodic properties of the signal. The problem with the very slow rate of change of My,Root could also have been solved by increasing the simulation time to take into account at least one rotation of the rotor. Due to the very slow rotor speed, the required simulation times would probably be in the range of one hour to several hours (possibly even 10 hours or longer, depending on how fast the rotor turns).

Nevertheless, the approach we chose (using 10-minute intervals) should also work, as we averaged out the influence of the very slow rate of change of My,Root over the high number of simulations and the initial start position of the rotor (random azimuth angle). For this reason, the determined run-in times should be correct. As you already described in your comment, this was of course not an efficient option and we agree that the simulation length is another factor that could/should be investigated. We therefore added this as an additional aspect for future work in the conclusions (see lines 431 to 436). However, when investigating this, one has to keep in mind two important aspects. First, especially if the simulation time required would be several hours, one would then have to ask whether this simulation is still realistic itself, since wind and wave conditions in reality do not usually remain constant for such a long time period. Thus, the simulation does not accurately reflect the reality, as the aeroelastic simulation usually assumes constant wind and wave conditions during the same simulation. Second, with a very long simulation time, it would be difficult to consider short idling periods occurred during the lifetime, as it would only be possible to consider periods within the required simulation length.

However, for the subsequent creation of the meta-model, the "short" simulation time of 10 minutes should not play a role, as both the initial starting position of the rotor and the average rotor speed are taken into account as input parameters in the process of the training of the meta-model. This takes into account the effect of the slow change of  $M_{y,Root,DEL}$ .

3) I wonder what is the total contribution from idling events to the lifetime fatigue damage accumulated on a typical site? This could give a very tangible insight into the importance of doing this type of study, and I believe it should be simple for the authors to estimate the number.

That is an important question. For the offshore wind turbine investigated in this work, a first investigation shows that the fatigue loads (lifetime DELs) are up to 40 % higher at 90 % availability compared to 100 % availability of the wind turbine. However, the results significantly depend on the type of turbine and considered location at the turbine, e.g., blade or monopile. We assumed that "available" means that the wind turbine is in normal operation for wind speeds between cut-in and cut-out wind speed and in idling operation for all other wind speeds. "Not available" means that the wind turbine is idling for all wind speeds. This aspect is in fact the subject of our current work. This is the reason, why a more detailed answer is not possible within this context, as a lot more information is needed to properly explain the results, which would go way beyond the scope of the work in the current paper.

For larger offshore wind turbines, the impact of the standstill conditions may be even higher as a study of Velarde et al. (2020) for a 10 MW offshore wind turbine on a monopile shows. They considered the wind speed range of power production assuming an availability of the wind turbine of 95 %. In this case, the parked/standstill conditions are responsible for 45 % to the total fatigue damage.

**Specific comments**

4) Abstract: I think the first few sentences in the abstract are unclear. I wouldn't call the surrogate model an alternative to aeroelastic simulations, since one needs to run aeroelastic simulations to train the model. The surrogate model is literally a surrogate or a substitution of the aeroelastic simulations when repeated runs are needed – in that sense, the surrogate models help us by making sure we only need to run the high-fidelity simulation experiment once.

Thank you for this comment. We agree with your point that the statement that metamodels are an alternative to simulations in the time domain is misleading. Because of this, we adjusted the first few sentences (based also on the comments from reviewer 1). We modified the beginning of the abstract to state that meta-models, as surrogate models for aeroelastic simulation models, are a good opportunity to perform lifetime calculations with a feasible computational effort (see lines 1 to 3).

5) Page 2, line 39: IEC 61400-1, ed4 is the current standard. It also considers idling, so I would cite the most recent standard.

Thank you for your comment. We updated the reference as you suggested (see line 44).

6) Page 2, lines 48-50: I think the use of "load cases" here is confusing. I believe the authors mean that approximately 5% of the operational lifetime (5% of the reference 10-minute periods) are spent under idling conditions. In IEC61400-1 lingo however, "load cases" are specific scenarios (Design Load Cases, DLCs) to be simulated, and they don't necessarily correspond to a specific time fraction. The "idling" DLC is one – DLC6.4. I suggest replacing "load cases" with "10-minute realizations" or similar.

Thank you for pointing that out. We appreciate that this may be confusing. We replaced the term "load cases" with "10-minute realisations", "combinations of environmental parameters" or similar terms, as you suggested.

7) Page 2, lines 54-55: while 5-15% of the lifetime indeed corresponds to more than 100,000 ten-minute periods, I wouldn't agree that we need 100,000 simulations to cover it as normally the load cases will create a lookup table for a few combinations of wind conditions and apply probability weighting. For example, the normal production load case DLC1.2 normally requires anything between 200 and 1000 simulations, depending on the turbine technology and onshore/offshore configuration. I don't see why we would need more for idling.

It is correct and we are aware that, in the design according to the IEC standards, a significantly lower number of 10-minute realisations is usually simulated compared to the high number mentioned in our paper (for both normal operation and idling). However, by simulating only a reduced number of, for example, 200-1000 input parameter combinations, and using probability weighting, not all input parameter combinations that really occur during the lifetime are taken account. As a result, this calculation can be only an approximation of the "real" lifetime, taking into account all possible input parameter combinations. We investigated this in a previous study

(Schmidt et al. (2023)), in which we analysed the influence of the number of simulations on the lifetime DEL in the approach according to the standard and compared it with a "full" lifetime calculation, for which 100,000 simulations (corresponding to approx. 1.9 years) were performed. This study clearly showed that the deviation of the calculated lifetime DEL according to the standard from the calculated lifetime DEL of the "full" lifetime calculation depends strongly on the number of input parameter combinations considered in the approach according to the standard.

When using the meta-models, we want to take advantage of the benefits of the meta-models by considering all combinations of environmental parameters that have actually occurred to calculate the "real" lifetime. This results in more than 100,000 combinations of environmental parameters that must be taken into account for idling.

To make this point clearer, in lines 62 to 63, we supplemented the sentence in which we mention the 100,000 simulations to make it clear that the 100,000 simulations are due to the fact that all actually occurred combinations of environmental parameters are taken into account.

8) Page 4, line 121-123: 10,000 simulations per wind speed to determine run-in times sounds excessive. With 16 wind speed bins, does this mean 160,000 simulations in total? And I believe the 2m/s step is not ideal if we are interested in the idling range, because we have most of the idling happening at very low wind speeds, and we may want higher resolution at those ranges.

Yes, you're right. We conducted 10,000 simulations per wind speed, resulting in 170,000 simulations in total because we considered 17 wind speeds. As already stated in our response to point 2), this could have been organised more efficiently. However, it is not the case that this effort has to be repeated every time simulations of another idling wind turbine have to be carried out. In this case, it should usually be sufficient to investigate whether the results for the run-in time in this work also apply to the new wind turbine type or whether the run-in time needs to be adjusted slightly.

A step size of 2 m/s is certainly not a suitable choice in certain cases, as many input parameter combinations are not covered correctly, especially if idling is only to be taken into account for very low wind speeds or very high wind speeds. However, we only used the step size of 2 m/s to determine the run-in times and we were interested in the whole wind speed range. Since we did not see a clear trend in this investigation that the step size should be defined more precisely, we assumed that the step size of 2 m/s is sufficient for this investigation. Of course, the investigation could also have been conducted differently, by simulating 10,000 random input combinations per wind speed bin, as in Hübler et al. (2017), rather than simulating the same input parameter combination 10,000 times for each wind speed. We discussed this point in the paper in lines 179 to 193.

Nevertheless, a step size of 2 m/s is not sufficient for meta-modelling. For this reason, as we described in our paper, we used a Halton sequence to generate the training and test data for meta-modelling in order to cover the parameter space optimally.

9) Section 3.2, line 238: another 10,000 simulations are used to train the Kriging model. Maybe the simulations done for determining the run-in time can be reused instead?

Unfortunately, this is not possible, as we have only simulated one input parameter combination for each wind speed to determine the run-in time. This means that we used a total of 17 different input parameter combinations. This number is not sufficient for

creating a meta-model that adequately covers the entire input parameter space. As mentioned in the previous point, the 10,000 additional simulations to train the Kriging model were created with a Halton sequence which was selected to achieve a sufficiently good coverage of the parameter space. During processing, it became apparent that significantly fewer (2,000) simulations would have been sufficient. However, in order to find this out, this high computational effort had to be carried out first.

**References used in responses to reviewers:**

Hübler, C., Gebhardt, C. G., Rolfes, R.: Development of a comprehensive database of scattering environmental conditions and simulation constraints for offshore wind turbines, Wind Energ. Sci., 2, 2, 491-505, 0.5194/wes-2-491-2017, 2017.

Müller, F., Krabbe, P., Hübler, C., Rolfes, R.: Assessment of meta-models to estimate fatigue loads of an offshore wind turbine, in: The Proceedings of the Thirty-First (2021) International Ocean and Polar Engineering Conference, Rhodos, Greece, 20-25 June 2021, 543-550, 2021.

Müller, F., Hübler, C., Rolfes, R.: Transferability of meta-model configurations for different wind turbine types, in: Proceedings of the ASME 2022 41st International Conference on Ocean, Offshore and Arctic Engineering, Vol.8: Ocean Renewable Energy, OMAE 2022, Hamburg, Germany, 5-10 June 2022, 79698, https://doi.org/10.1115/OMAE2022-79698, 2022.

Santos, F. d. N., Noppe, N., Weijtjens, W., Devriendt, C.: Results of fatigue measurement campaign on XL monopiles and early predictive models, Journal of Physics: Conference Series 2265, 032092, 10.1088/1742-6596/2265/3/032092, 2022.

Schmidt, F., Hübler, C., Rolfes, R.: Lifetime reassessment of offshore wind turbines using meta-models, in: 14th International Conference on Applications of Statistics and Probability in Civil Engineering (ICASP 14), Dublin, Ireland, 9-13 July 2023, 2023.

Velarde, J., Kramhøft, C., Sørensen, J. D., Zorzi, G.: Fatigue reliability of large monopiles for offshore wind turbines, Int. J. Fatigue, 134, 105487, 10.1016/j.ijfatigue.2020.105487, 2020.

Ziegler, L., Schulze, H., Henning, M.: Optimization of curtailment intervals of wind turbines through assessment of measured loads during start-up and shutdown events, Journal of Physics: Conference Series 2767, 032006, 10.1088/1742-6596/2767/3/032006, 2024.